# A Network Approach for Multiscale Catchment Classification using Traits

Fabio Ciulla[1] and Charuleka Varadharajan[1]

[1]Earth and Environmental Sciences Area, Lawrence Berkeley National Laboratory, Berkeley, CA

**Correspondence:** Fabio Ciulla (fciulla@lbl.gov)

**Abstract.** The classification of river catchments into groups with similar biophysical characteristics is useful to understand and predict their hydrological behavior. The increasing availability of remote sensing and other large-scale geospatial datasets have enabled the use of advanced data-driven approaches to classify catchments using traits such as topography, geology, climate, land cover, land use, and human influence. Unsupervised clustering algorithms based on the Euclidean distance are commonly used for trait-based classification, but are not suitable for high dimensional data. In this study we present a new network-based method for multi-scale catchment classification, which can be applied to large datasets and used to determine the traits associated with different catchment groups. In this framework two networks are analyzed in parallel; the first where the nodes are traits, and the second where the nodes are catchments. In both cases, edges represent pairwise similarity and a network cluster detection algorithm is used for the classification. The traits network is used to investigate redundancy in the trait data and to condense this information into a small number of interpretable categories. The catchments network is used to classify the catchments into clusters, and to identify representative catchments for the different groups using the degree centrality metric. We apply this method to classify 9067 river catchments across the contiguous United States at both regional and continental scales using 274 non-categorical traits. At the continental scale, we identify 25 interpretable trait categories and 34 catchment clusters of size greater than 50. We find that catchments with similar trait categories are typically located in the same region, with different spatial patterns emerging among clusters dominated by natural and anthropogenic traits. We also find that the catchment clusters exhibit distinct hydrological behavior based on an analysis of streamflow indices. This network approach provides several advantages over traditional means of classification including better separation of clusters, the use of alternate similarity metrics that are more suitable for high dimensional data, and reducing redundancy in the trait information. The paired catchment-trait networks enables analysis of hydrological behavior using the dominant trait categories for each catchment cluster. The approach can be used at multiple spatial scales, since the network topologies adjust automatically to reflect the trait patterns at the scale of investigation. Finally, the representative catchments identified as hub nodes in the network can be used to guide transferable observational and modeling strategies. The method is broadly applicable beyond hydrology for classification of other complex systems that utilize different types of trait datasets.

# 1 Introduction

Catchments are complex environmental systems that consist of diverse natural and anthropogenic components interacting non-linearly in space and time. A fundamental challenge in hydrology is to understand how these interacting components influence critical catchment functions such as stream flow and solute exports (Troch et al., 2015). This is difficult to determine even for catchments with little to no human influence because of the spatial heterogeneity of processes across regions with different characteristics leading to the problem of "uniqueness of place" (Beven, 2000). For the majority of catchments, this challenge is

compounded due to anthropogenic activities that cause more complex behavior that are not easily modeled (Sivakumar et al., 2015; Sivapalan, 2006).

The classification of river catchments into groups with similar characteristics has been called out as a practical approach to address the diversity of hydrologic behavior (McDonnell and Woods, 2004; McDonnell et al., 2007; Wagener et al., 2007). A classification system would establish the baseline for similarities and differences among catchments, and would be extremely

beneficial for both modeling and experimental analysis (Dooge, 1986). Several regionalization approaches that attempt to transfer knowledge (observations, theory, or model predictions) from well observed sites to other less-observed regions, such as those used for predictions in ungaged basins (PUBs), rely on the concept of similarity for extrapolation (Guo et al., 2021; Merz and Blöschl, 2004). The ultimate goal of a classification process is to discover the laws governing the behavior of a system under investigation. By looking at the collective behavior of a group of similar elements, the unique contributions from

individual elements average out, revealing the fundamental characteristics that regulate catchment functions (Sokal, 1974).

There is a long history of using physical characteristics and other properties (i.e., their traits) to classify catchments in hydrology, as described in Wagener et al. (2007). The simplest approach uses dimensionless numbers or indices such as stream order (Horton, 1945; Strahler, 1952) and Peclet numbers (Redner, 2001) as organizing constructs. More complex approaches use distribution functions (e.g. hypsometric curve; Langbein (1947)), and conceptual or mathematical models. A common

approach is to group catchments by hydroclimactic region such as the Koppen classification (Koppen, 1918) or the Budyko framework (Budyko, 1974). Many studies use the dynamic behavior of a catchment as quantified by the signatures of a function of interest as the basis for classification instead of traits (see references in McMillan (2020)), and find the results to be consistent with knowledge about watershed processes (McMillan et al., 2022). However, signature-based classification cannot be used for unmonitored sites, which is a significant issue given the paucity of stream flows and other hydrological measurements.

There has been a dramatic increase in the amount of regional to global scale geospatial datasets produced over the past two decades, which has enabled a new era of data-driven hydrology (Hubbard et al., 2020). A few data products such as the Geospatial Attributes of Gages for Evaluating Streamflow, version II (GAGES-II; Falcone (2011)), StreamCat (Hill et al., 2016), Caravan (Kratzert et al., 2023) and Hysets (Arsenault et al., 2020) now provide extensive information on hundreds of traits such as land surface structure, climate conditions, vegetation, land use and other human influences across thousands of

catchments. Thus new methods for classification that go beyond simplistic trait representations to account for a diverse array of natural and anthropogenic catchment properties are now possible, and would be extremely useful to understand or predict complex system behavior. A classification approach would use the trait features as vectors in a multidimensional space and the

relationships among the catchments defined by a distance metric on this space. This approach has been used in recent attempts to group regions with unsupervised machine learning algorithms such as K-means and hierarchical clustering (Sawicz et al., 2011; Wainwright et al., 2022; Kumar et al., 2011).

However, the analysis of large multivariate datasets has two main challenges that need to be addressed. The first is multicollinearity, which is the possibility of information redundancy in the data due to the presence of multiple variables that provide similar information. Machine learning algorithms for classification generally have degraded performance when multicollinearity is present. The second issue is the "curse of dimensionality" (Bellman, 2010), a phenomenon that emerges when using distance metrics to compute the relationship amongst data points represented as vectors of features (e.g., traits) in a multidimensional space. When the number of dimensions increases, the density of data points drastically decreases making the feature space more sparse (Houle et al., 2010), and causes the difference between the furthest and the closest distance to a point to approach zero (Beyer et al., 1999). This results in degraded performance of similarity measures, and in particular some metrics such as the commonly used Euclidean distance are more affected than others (Aggarwal et al., 2001).

Many algorithms typically used for unsupervised classification, including machine learning methods such as K-means and hierarchical clustering do not address the dual issues of multicollinearity and data dimensionality. The K-means algorithm (MacQueen, 1967) is widely used across many fields and relies on the establishment of cluster centroids in a vectorial space that act as representative points for the clusters. The data points are associated with a cluster by minimizing their Euclidean distance, based on the variances within the clusters. However, as the dimension of the vector space grows, the Euclidean distance becomes unreliable to properly quantify the relationship among the data points (Aggarwal et al., 2001). Additionally, the Euclidean distance discards information about the directionality of the data, producing a value that reflect only the relative position of data points in the vector space, and neglecting the contribution of the different components that can be relevant for classification. Another group of algorithms for cluster identification are referred to as agglomerative hierarchical clustering. This class of methods aims to build a hierarchy amongst the data points, and different strategies can be used to find the optimal partitioning. Some of the most notable methods are the complete and single linkage clustering, which respectively aim to maximize or minimize the distance of two points belonging to a different cluster (Müllner, 2011). Although these two methods do not necessarily rely on the Euclidean distance, they are quite sensitive to outliers because they are based on the extreme (i.e., maximum or minimum) distances between points. Such dependence on single data points for partitioning goes against one of the main concepts in the characterization of complex systems, where emergent properties (i.e., the grouping into clusters), are the result of a collective phenomena rather than the disproportionate contribution of a single element. The Ward's algorithm (Ward, 1963) mitigates the issue of outliers by introducing the minimum variance method, but is based on the Euclidean distance and therefore subject to the same drawbacks as the K-means method.

Approaches based on network science have been used to investigate complex systems for many applications involving large datasets (Newman, 2018; Börner et al., 2007). The ability of networks to capture complex interactions among different parts of a system and to highlight emergent behaviors has been proven to be extremely powerful (Strogatz, 2001; Boccaletti et al., 2006). By treating each element as a node, and the relationship amongst them as edges, the structure of a network provides a natural and intuitive representation of a complex system (Barrat et al., 2004; Albert and Barabási, 2002). Modeling the relationships

amongst elements of a complex system as a network also enables the use of suitable distance metrics for large datasets, presenting a solution to address the issues of data dimensionality and multicollinearity. Networks are used extensively in many different scientific domains. For example, in social science, political polarization has been detected with the presence of clusters in a social network (Conover et al., 2011). In biology, protein-protein interaction networks are essential to understand the cell physiology (Jeong et al., 2001), and plant trait networks are used to understand plant adaptation to different environments by examining the complex relationships between their functional traits (He et al., 2020). In hydrology, the concept of networks has mainly been associated with the tree-like structure of rivers, and has enabled advances in understanding geomorphological processes (Rinaldo et al., 2006; Rodríguez-Iturbe and Rinaldo, 2001; Tejedor et al., 2017; Czuba and Foufoula-Georgiou, 2014). The use of networks as a generic framework for hydrological investigation has been postulated (Sivakumar, 2015) and the concepts of complex flow networks have been used to understand river transport in deltas (Sivakumar et al., 2015; Tejedor et al., 2018). However, to date, network theory has not been used for understanding relationships between catchments or their traits.

In this study, we introduce a novel, network-based method to classify catchments at multiple scales based on traits representing their climatic conditions, vegetation, topography, soils, land use, and anthropogenic characteristics. This unsupervised method utilizes two parallel networks to extract clusters of similar traits and catchments, while using dimensionality reduction and a cosine distance metric, to address the issues of information redundancy and high dimensionality present in large geospatial datasets. This approach enables characterization of catchments using a small number of interpretable trait categories, and provides a more generalizable approach to regionalization of hydrologic behavior. We demonstrate the utility of our network-based method for analyzing catchment properties and associated hydrological behavior for 9067 catchments and 274 traits across the continental United States. To our knowledge, this is the first instance where networks have been used for trait-based catchment classification in the hydrological sciences. Our methods are broadly applicable beyond catchment classification for analysis of other environmental traits (e.g., plant functional traits or microbial traits) and large datasets.

## 2 Methods

In this section, we describe the dataset and the workflow for unsupervised trait-based classification of watersheds using the network approach. The steps in our classification workflow (Fig. 1) include: (1) downselection and preprocessing of traits from the GAGES-II dataset (Sect. 2.1) by transforming and standardizing trait values (Sect. 2.2), (2) removal of redundant information and multicollinearity using the principal component analysis (PCA), which produces low dimensional vector representations of both traits and catchments (Sect. 2.3), (3) computation of the pairwise similarities of the dimensionally-reduced vectors of both catchments and traits (Sect. 2.4), (4) generation of two similarity networks, one where nodes are catchments and another where nodes are traits (Sect. 2.5). In both cases edge weights represent the pairwise similarities within the respective sets, (5) classification via a clustering algorithm to identify trait categories and groups of catchments with similar traits (Sect. 2.6), (6) characterization of the catchment clusters using the over and under expression of traits categories (Sect. 2.8), (7) evaluation of the spatial coherence between catchment clusters (Sect. 2.7), (8) selection of representative catchments using network central-

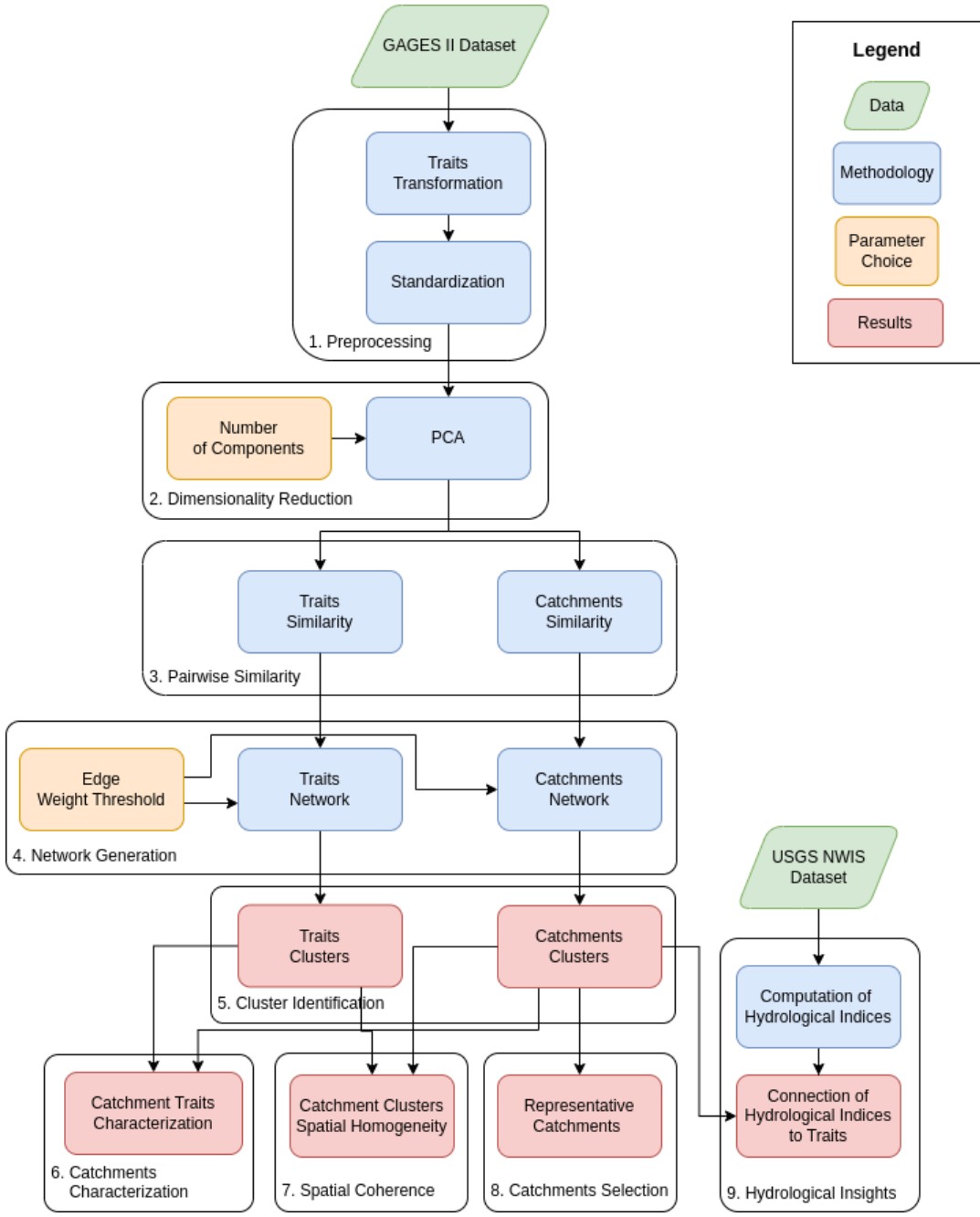

**Figure 1.** A schematic of the workflow used in this study.

ity measures (Sect. 2.9), (9) identifying hydrological behavior based on streamflow indices across the catchment clusters and association with their characteristic traits expression (Sect. 2.10).

## 2.1 Datasets

The primary dataset used in this study is the Geospatial Attributes of Gages for Evaluating Streamflow, version II (GAGES-II), as described in Falcone (2011). This dataset contains a comprehensive set of geospatial characteristics for 9,322 gaged catchments in the contiguous United States (CONUS), as well as Alaska, Hawaii, and Puerto Rico with long flow records (at least 20 complete years of discharge record since 1950) or having an active record as of water year 2009. Here, we refer to 'catchments' as the area upstream of each individual gaging station in the GAGES-II dataset, of which 2,057 are considered pristine ('reference') and 7,265 are disturbed by human influences ('non-reference').

The geospatial attributes, referred to as 'traits' in this paper, are compiled from various data sources for the CONUS (see Falcone et al. (2010) for details). The 354 traits in the dataset span climate (e.g. historical average precipitation and air temperature), soil types and composition, geomorphology (e.g. topography), vegetation (e.g. extent of forests), surface waters (e.g. extent of lakes and wetlands), stream characteristics (e.g., sinuosity and stream order), and anthropogenic influences (e.g. proximity to developed areas, land use or presence of dams). Some traits have multiple representations; for example, elevation is provided at the location of the stream gage and as an average across the catchment. Many of the climate traits are calculated from the 800 m gridded PRISM dataset (Daly et al., 2000), which is derived from 30-year records that span different time periods (primarily 1961-1990 or 1971-2000). We highlight that the runoff variables present in the dataset are not measured values of the discharge, but instead are estimates from 1951-2000 computed by a water balance model for 4 km grids using precipitation and temperature as inputs (Wolock and McCabe, 1999). Land use and land cover data are derived from the 2006 USGS National land cover dataset. We selected the GAGES-II dataset for our unsupervised classification because of its comprehensiveness across catchments, diversity of traits and spatial coverage.

The second dataset used in this study consists of streamflow data obtained from the USGS National Water Information Service (USGS, 2022) for the period 1971 to 2000. We chose this time window to overlap with the period of data availability for most of the traits used from GAGES-II.

## 2.2 Data Filtering and Preprocessing

This study focuses on the 9067 stream gages from catchments that are present within the CONUS in the GAGES-II dataset, excluding the gages in Alaska, Puerto Rico and Hawaii. Out of the 354 traits present in the dataset, we selected 274 that had numerical values and discarded 74 categorical variables (e.g., geology type) or textual labels (e.g., county name). We also discarded 6 additional traits: station identification number (STAID), the 2 digit hydrological unit (HUC02; Seaber et al. (1987) that each station belongs to, latitude (LAT_GAGE) andlongitude (LNG_GAGE) of the station, soil with variable drainage characteristics (HGVAR) and mean watershed aspect (ASPECT_DEGREES). The STAID, HUC02, LAT_GAGE and LNG_GAGE variables contain explicit information about the location of the gaging stations. We chose to exclude them to develop the classification scheme solely using natural and anthropogenic features, which avoids bias due to spatial proximity and is transferable to any location. Additionally, their exclusion for the classification enables an unbiased evaluation of the emerging spatial patterns amongst the resulting catchment clusters. The HGVAR variable is removed because it has zero variance and thus cannot

be standardized in the preprocessing routine. The ASPECT_DEGREES trait is removed because its periodicity has poor physical coherence (both the quantity 0 and 360 convey the same information but are represented by the extreme values of the variable range). Furthermore the aspect information is present in two other variables that we include, ASPECT_EASTNESS and ASPECT_NORTHNESS, which are the sine and cosine of the ASPECT_DEGREES variable respectively.

The preprocessing step is composed of two operations (step 1 in Fig. 1). The first involves feature transformations of non-monotonic traits to a consistent monotonic function. For example, several traits denote the presence of points of interest in the basin such as dams and canals. However, when such a point is not present in the basin, the trait is assigned the value -999. This creates an inconsistency with the physical distance values, and thus is remapped to be physically coherent for use as an input in the PCA method. Further details on mapping of variables to monotonic functions are in Appendix A. The second step is

data standardization, which involves subtracting the arithmetic mean from each variable and dividing the result by the standard deviation. As a result, the standardized values will have zero mean and standard deviation equal to 1. The standardization is essential for the PCA, which requires the data to have a zero mean, and additionally scales variables in the dataset that span up to five orders of magnitude into a comparable set of values and variances needed for the PCA projections. Each of the 9067 catchments is represented as a vector of these filtered and remapped 274 traits. We provide the GAGES-II dataset preprocessed

according to these steps as part of the dataset (Ciulla and Varadharajan, 2023).

## 2.3   Dimensionality Reduction

The next step in the workflow (Step 2 in Fig. 1) involves reducing the dimensionality of the pre-processed dataset using the Principal Component Analysis (PCA) algorithm (Pearson, 1901), which is required for two reasons. First, it is used to mitigate the issue of the *curse of dimensionality*, which results in reduction of performance of distance metrics in high-dimensional

vector spaces such as the 274-dimensional trait vectors used to represent catchments in our study (Aggarwal et al., 2001). Secondly, the dimensionality reduction minimizes the redundancy in the dataset caused when information present in one trait is also represented (partially or totally) in other traits. Information redundancy is generally detrimental for machine learning algorithms and needs to be addressed. This information redundancy can be detected by computing the trait correlations and is referred to as multicollinearity. The traits in the GAGES-II dataset contain significant redundancies, with 84% of pair-wise

Pearson correlation coefficients (Pearson, 1895) and 92% of pair-wise Spearman coefficients, which accounts for non-linear relationships (Spearman, 1987), have a significant p-value of 0.05. The coefficient of determination between these two metrics is equal to 0.76, which indicates that although nonlinear relationships among the traits are present, they are not so dominant to prevent the use of a linear dimensionality reduction method such as PCA. Another factor that can affect the PCA algorithm's performance is the presence of outliers. We determined that the PCA is a reasonable choice for the GAGES-II dataset, since

only 8.1% of the traits lies outside their "inner fence", a common threshold for outliers, defined as the range between Q1 - 1.5 * IQR and Q3 + 1.5 * IQR for each trait, where Q1 and Q3 are the first and third quartiles respectively, and IQR = Q3 - Q1 is the interquartile range.

    The PCA algorithm collapses the dimensions of linearly correlated variables into a lower dimensional vector representation. The information of the variables is retained in directions called principal components that maximize the variance of the original

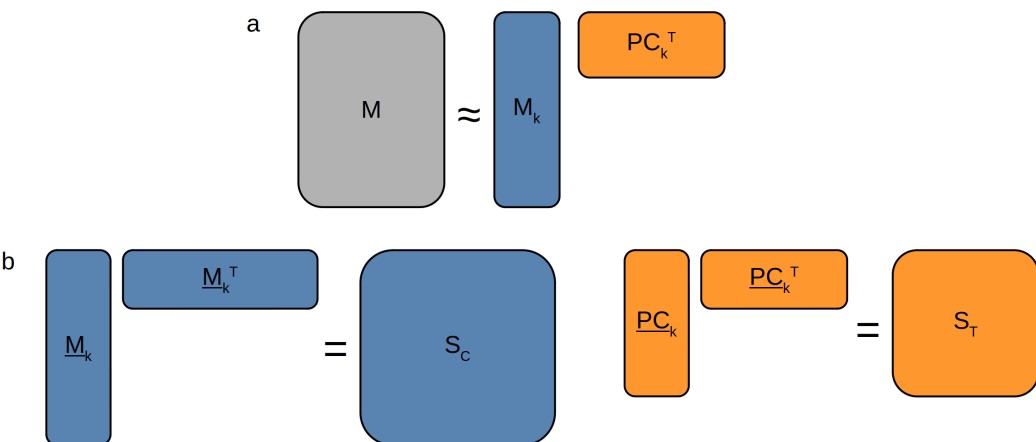

**Figure 2.** Pictorial representation of (a) Dimensionality reduction via PCA of the original matrix $M$ into the transformed matrix $M_k$, encoding information about catchments and the matrix of principal components $PC_k$, encoding information about the traits, and (b) The generation of the catchments similarity matrix $S_C$ from the L2-normalized $\underline{M}_k$ and the traits similarity matrix $S_T$ from the L2-normalized $\underline{PC}_k$.

data, thus addressing the issues of multicollinearity and high dimensionality. In the PCA, the dataset is represented as a matrix $M \in \mathbb{R}^{n \times f}$ where $n$ is the number catchments and $f$ is the number of traits, and each catchment is represented by a multidimensional vector of traits in the $f$-dimensional space (Fig. 2). The PCA transforms the $n \times f$ matrix $M$ into a new $n \times k$ matrix $M_k$, where the catchments are represented by $n$ low dimensional vectors of size $k$. The only free parameter of the algorithm is the number of final dimensions $k$, which reflects the amount of information retained. By using the Cao's implementation (Cao,
1997) of the false nearest neighbors (FNN) method (Krakovská et al., 2015), we set $k = 20$ (see Appendix B for details).

Hence, the original matrix $M$ is transformed into the matrix $M_k \in \mathbb{R}^{n \times k}$ containing 9067 vectors of dimension 20, where each vector is a low-dimensional representation of the traits in each of the catchments, which is used in the following steps of our analysis. Complementary to $M_k \in \mathbb{R}^{n \times k}$, the matrix of principal components $PC_k \in \mathbb{R}^{f \times k}$ contains $f = 274$ vectors of size $k$ that encodes information about how the traits are expressed in the catchments. The analysis of $PC_k \in \mathbb{R}^{f \times k}$ is used to
identify the relationships and information redundancy amongst the different traits. To summarize, the PCA results in a set of $n$ $k$-dimensional vectors that encodes the trait information for each catchment, and $f$ $k$-dimensional vectors that describe the traits expressions.

## 2.4 Similarity Measure

We identify the relationships between catchments and traits using the cosine similarity as a distance metric. Cosine similarity is defined as in Eq. 1 (Salton, 1983):

$$S_C(\boldsymbol{x}, \boldsymbol{y}) = \frac{\boldsymbol{x} \cdot \boldsymbol{y}}{\|\boldsymbol{x}\| \|\boldsymbol{y}\|} \tag{1}$$

where $x$ and $y$ are two vectors with the same lengths. The cosine similarity is preferred over the classical Euclidean distance metrics for two reasons. First, the reduced-order vectors from the PCA are still considered to be high-dimensional data for distance calculations. Secondly, the cosine distance retains information about directionality of the data (see Appendix C for additional explanation about the rationale for the choice of this similarity measure).

The cosine similarity is computed as the dot product of the L2 normalized matrices $M_k$ and $PC_k$ with their corresponding transposed matrices (Fig. 2b). This results in two matrices $S_C$ and $S_T$ of size $n \times n$ and $f \times f$ that contain the pairwise similarities between catchments and traits respectively. The diagonal elements of both matrices are unit values because they are the results of the dot product of normalized vectors with themselves.

## 2.5 Network Generation

In this study, we adopt networks as a tool to investigate the relationship among the catchments. For more details on network theory see Appendix D. We use the cosine similarity matrix $S_C$ to build a network $N_C$ where nodes are catchments and the edges represent their similarity (Step 4 in Fig. 1). In particular, the edge weights are the cosine similarity values mapped to a range between 0 and 1 (Equation 2):

$$A = \frac{S_C + 1}{2} \tag{2}$$

Thus, two nodes are connected if they are similar as per the cosine metric, and the strength of the similarity is reflected by the edge weights. The self-loops (i.e., an edge that originates from, and ends at the same node) given by the unitary values of the diagonal similarity matrix are deliberately discarded. The connectivity patterns in the network reveals clusters of nodes that are more similar to each other than to the rest of the network.

By assigning an edge to each pair of nodes using the similarity metric we produce a fully connected network, namely a network with all the possible $N_C(N_C - 1)/2$ edges. Such a network is a relatively uninteresting one, because the presence of all possible edges hides eventual complex topological patterns present in it. In order to reveal such patterns, we apply a filtering mechanism to our set of edges. In particular, we use the disparity filter as described in Serrano et al. (2009). Such a method takes a network as an input and filters out some of its edges returning what is called the network backbone. The disparity filter assumes complete homogeneity among the edge weights with the null hypothesis that the weights of the edges incident to a node are uniformly distributed. Then it retains only the edges whose weight magnitude is incompatible with such an hypothesis according to a certain significance level. By repeating this process for each node in the network, we generate the network backbone, where all the edges are statistically validated against the uniform distribution hypothesis. Only edges

are affected by this technique, while the nodes are unaffected and their number in the network stays the same. The method does not prescribe a universal value for the significance level, which has to be evaluated case by case. In our implementation of the disparity filter, we choose a significance level where 95% of the nodes in the network belong to the giant component, which is the connected component in a network that includes most of the nodes. The choice of 100% of the nodes in the giant component is impractical because there could be nodes that potentially never connect to any other ones, making the 100% value impossible to reach.

The procedure used for generating the catchments network is also used to produce a network of traits $N_T$ via the matrix $S_T$ where each node is a trait and the edges connect two traits, with weights representing how similarly they are expressed across the catchments. Similar to the catchment case, we extract the backbone network to reveal complex patterns among groups of traits.

For catchments, the backbone network $N_C$ contains 9067 nodes and 559207 edges. The number of edges corresponds to only 1.4% of the ones that can possibly be present in a fully connected network of such a size. In the case of the traits, the backbone network $N_T$ has 274 nodes and 1422 edges, comprising 3.8% of all the possible edges. These networks are used in the following step to identify clusters of similar catchments and traits.

## 2.6 Clusters Identification

A complex network is a network with non trivial connectivity patterns (Strogatz, 2001; Boccaletti et al., 2006). One of the main hints of the presence of such patterns is given by the heterogeneous distribution of the values of the clustering coefficient, a topological measure of the tendency of a network to form groups of nodes that are more connected to each other than with the rest of the network (Wasserman and Faust, 1994; Scott and Carrington, 2023). Appendix E contains a detailed discussion on clustering coefficients. These groups of nodes are called clusters and there are a variety of algorithms to discover them (Fortunato, 2010).

To obtain the division of nodes into clusters, we implement the Infomap clustering algorithm (Rosvall and Bergstrom, 2008) for both the catchment and the trait networks. This is an agglomerative clustering method for networks that starts by assigning each node to its own cluster and uses the map equation as described in Rosvall et al. (2009) to optimize the iterative merging of these partitions. One advantage of using the Infomap algorithm is its capability of splitting already formed clusters at successive iterations, if this action benefits the clustering optimization function. When applied to the catchment network, 95% of the nodes in the entire network are contained in the top 71 clusters ranked based on their size. We provide the number of clusters at 95% coverage because, as usual for complex networks, the cluster size spans several orders of magnitude, and thus accounting for all the clusters is inappropriate. The cluster size ranges from 1 to 953 nodes, with the latter comprising about 10% of the nodes of the entire network. Because of how the catchment network is built, catchments within a cluster have more similar traits than with the rest of the network. When the Infomap algorithm is applied to the traits network, the number of top clusters in size that comprises 95% of the network is 20 and their sizes range from 1 to 25. For this network, the clusters denote groups of traits that have similar expression patterns across the catchments.

The cluster identification represents the first milestone of our workflow (Step 5 in Fig. 1), resulting in classification of both traits and catchments using a completely unsupervised methodology. The only choices we made were informed by statistical analysis to set the two free parameters of the framework, namely the number of dimensions for the PCA algorithm and the confidence level for the backbone disparity filter.

## 2.7 Spatial Homogeneity of Catchment Clusters

Contrary to supervised methods that compute model performance using testing data, unsupervised algorithms like ours do not have a direct means of assessing their performance. Thus, to identify a measure of performance of our unsupervised catchment classification, we use the principle of Tobler's first law of geography, which states that "everything is related to everything else, but near things are more related than distant things" (Tobler, 1970). This law, applied to our classification analysis, would mean that there is a higher probability of catchments that are geographically close to each other belonging to the same cluster. To quantify this concept we compute the global homogeneity measure $H_{global}$ as the average probability that for each catchment the nearest neighbor in terms of spatial proximity belongs to the same cluster. This quantity can be interpreted as geographical homogeneity in the sense that catchments surrounded by others within the same cluster make the system more homogeneous.

$$H_{global} = \frac{1}{N_C} \sum_i h_i \tag{3}$$

and

$$h_i = \begin{cases} 1 & \text{if } c_i = c_j \\ 0 & \text{otherwise} \end{cases} \tag{4}$$

Here $N_C$ is the total number of catchments, $j$ is the closest catchment to catchment $i$, $c_i$ and $c_j$ are the clusters that catchments $i$ and $j$ belong to. $H_{global}$ provides a single value for the entire set of catchments, where values closer to 1 indicate that the system is more homogeneous. Additionally, we compute a within cluster homogeneity measure, $H_{cluster}^i$, as the probability of finding catchments belonging to cluster $i$ within the convex hulls $A_i$ of all catchments in $c_i$. This metric is similar to the relative abundance of species in ecology, and is computed for each cluster as the ratio between the number of catchments $N_A^i$ belonging to $c_i$ in convex hull $A_i$, and the overall number of catchments $N_A$ within this area:

$$H_{cluster}^i = \frac{N_A^i}{N_A} \tag{5}$$

If a convex hull contains only one type of catchment cluster, which by default will be the cluster that generates the convex hull, then the value of $H_{cluster}^i$ is equal to 1, indicating complete homogeneity.

## 2.8 Characterization of Catchments Clusters

One of the goals of this study is to provide a methodology that enables the interpretability of the resulting catchment classification. This would correspond to answering the question: what are the characteristic traits of the catchments belonging to

a particular cluster? We answer this question by identifying the set of traits that are over or under expressed amongst all the nodes of a particular catchment cluster (Step 6 in Fig. 1).

This is done by computing the z-score of each trait for each catchment in the dataset. This value represents the unique expression of a trait in a specific catchment. Next, the z-score values are aggregated for each catchment cluster as an arithmetic average producing a mean z-score of each trait for each catchment cluster:

$$z_{uv} = \frac{1}{\|c_v\|} \sum_{w \in c_v} \frac{x_{uw} - \mu_u}{\sigma_u} \tag{6}$$

Here $z_{uv}$ is the mean z-score of the trait $u$ in the cluster $c_v$ of size $\|c_v\|$. The value $x_uw$ is the value of the trait $u$ for the node $w$ in the cluster $c_v$. $\mu_u$ and $\sigma_u$ are the average and standard deviation of the trait $u$ in the entire dataset.

In this way, a high absolute value of the average z-score represents a substantive expression of a trait in a particular catchment cluster, with positive values indicating overexpression and negative values denoting underexpression of the trait.

However, investigating the unique expression of each of the 274 traits for each catchment cluster would disregard the redundancy of information identified by the trait clusters and neglect eventual collective contribution of several traits. Instead, a better analysis of trait expression can be achieved by considering their z-score values within the context of other traits in the dataset, by leveraging the clusters produced by the trait similarity network. So, instead of investigating the expression of each trait in isolation, we aggregate the z-scores of all the traits belonging to each cluster in the network of traits. In this way we characterize each catchment cluster by the expression of a few, easily interpretable groups of traits (referred to as trait categories). The advantage of the investigation of catchment clusters in the context of the network of traits will become more evident when we discuss the results in Sect. 3.

## 2.9 Identification of Representative Catchments

One advantage of modeling a system as a network is the ability to use a multitude of metrics from graph theory (Newman, 2018; Börner et al., 2007). One of the most used ones is the degree centrality (Börner et al., 2007), defined as the number of edges incident to a node.The nodes at the other end of these edges are referred to as neighbors. The interpretation of the degree varies according to the nature of the edges but, in general, nodes with a high degree play a more important role in the network and are often referred to as hubs. In the case of a similarity network, a node with a high degree centrality identifies an element that is very similar to many others, while a low degree node is similar to fewer nodes, and zero degrees indicates that the node is disconnected from the rest of the network. High degree nodes in similarity networks are the best candidates to be selected as representative nodes because they have the highest number of neighbors. By sorting the nodes according to their degree centrality, we can select a small number that are representative of the entire network.

We apply this concept to the network of catchments to select a small set that are representative of the cluster they belong to (Step 8 in Fig. 1). This is done by ranking catchments within each cluster and selecting nodes based on the degree centrality measure. However, we do not simply choose the nodes that have the highest degree centrality, since that could potentially lead to selection of two or more nodes that are topologically close to each other but both share a high number of neighbors, thus

undermining network coverage and resulting in redundant choices of representative catchments. Thus, to promote network coverage and minimize superfluous representation, our method first selects the node with the highest degree centrality in a cluster, and then removes both the selected node and its neighbors from the cluster. The new degree-based ranking for the remaining nodes is then updated due to the removal of nodes. We continue this selection process until the size of the union set of representative catchments and their neighbors is equal to $95\%$ of the size of the cluster they are extracted from. Using this methodology, nodes with an initially high degree centrality, which share many neighbors with a selected catchment (i.e., is topologically close from a network perspective), will have their degree reduced and are less likely to be selected.

## 2.10   Analysis of Streamflow Indices

The unsupervised classification workflow developed in this study finds groups of similar catchments using only traits, and does not use observations of hydrological variables such as stream flows or temperature. Hence, we can use the classification to examine the hydrologic behavior of the catchment clusters and their associated trait expressions. Here, we focus on streamflow regimes following the method in Olden and Poff (2003). They recommend using 34 representative flow indices, which are considered to minimize collinearity based on a review of 171 indices calculated using long-term flow records from 420 sites from across the CONUS. The list of these indices can be found in the Table S1 in the data repository associated with this paper (Ciulla and Varadharajan, 2023).

To compute the indices, we use the historical record of river mean daily discharge values for the 9067 gaging stations, retrieved using the BASIN-3D software (Varadharajan et al., 2022), which synthesizes timeseries data on-demand from different data sources including the USGS National Water Information System (NWIS) (USGS, 2022). We only consider the discharge data from 1971 to 2000 to match the time range of most of the traits in the GAGES-II dataset. We eliminate stations with more than 50% of missing values in our time range, resulting in a downselected set of 5251 stations. To compute the 34 hydrologic indices, we use the *EFlowCalc* package provided by Thibault (2021) that discards missing values in the computation of aggregated measures such as averages or standard deviations.

We aggregate indices as average values for each catchment cluster, after further downselecting to stations that are within 34 clusters containing at least 50 catchments. On average, 59% of the catchments in each of these 34 clusters have a corresponding long-term flow record for which we computed the hydrological indices. Given this relatively high percentage, we use the averaged streamflow indices from these catchments as proxies for the hydrological behavior of the entire cluster.

For all the hydrological indices, we use the statistical Kruskal–Wallis test (Kruskal and Wallis, 1952) to determine if there is a statistically significant difference between the distribution of the values between individual clusters and the rest of the catchments. This test does not assume normality in the data distribution and is the non-parametric alternative to the ANOVA method, which are necessary considerations because none of the streamflow index values were normally distributed as per the Shapiro test (Shapiro and Wilk, 1965) . The result of the Kruskall-Wallis test indicates if the indices grouped according to the clusters are drawn from the same distribution as the entire set. If that is not the case, it means that catchment clusters resulting from our trait-based classification have different streamflow characteristics.

We conducted two additional statistical tests to further examine whether the hydrological indices of catchment clusters are significantly distinct. The first is a nonparametric 1-sample Kolmogorov-Smirnov (K-S) Test that compares a sample distribution to a reference one for each hydrological index. This expands on the Kruskall-Wallis test, but allows us to determine the number of clusters that are statistically distinct from the entire catchment dataset. Here, each sample is constituted by the indices of one cluster and the reference distribution is based on all the catchments within the CONUS. The null hypothesis is

that samples are drawn from the reference distribution when using 0.05 as threshold for the p-value. The second is a 2-sample K-S test comparing the distributions of indices for pairs of catchment clusters, which allows us to determine how different the clusters are from each other. Here, each sample pair is constituted by the distribution of indices for the clusters being compared. Similar to the one-sample test, the null hypothesis is that samples are drawn from the same distribution when using 0.05 as threshold for the p-value.

## 375   2.11   Comparison with Traditional Clustering Techniques

We compare the clusters obtained from our methodology that uses network theory and the cosine distance metric, with the ones resulting from the traditional K-means and hierarchical clustering approaches. Since these are all unsupervised methods, we cannot use any target variable to compute the accuracy of the classification to compare performance. For this reason, we identified two metrics to evaluate the performance of the different methods.

The first metric, which we refer to as "cluster similarity", reflects the similarity between traits of the catchment clusters, which are represented by the average trait z-scores aggregated across the catchments in each cluster as described in Sect. 2.8. Here, each catchment cluster is compared to the others by calculating their pairwise cosine similarity. The highest value of the cosine distance within each catchment cluster is used as a conservative measure of inter-cluster similarity, to assess how far apart the catchment clusters are from each other. The median value of the inter-cluster similarities represents how distinct the

clusters produced by each algorithm are. We aim to minimize this metric, since a good classification algorithm should produce more distinct clusters.

    The second metric is the silhouette score (Rousseeuw, 1987), which is a measure of intra-cluster similarity. It represents how similar each element (i.e., a catchment) is to other elements within its cluster relative to elements in other clusters. The values of this metric range between -1 and 1, with higher values denoting that an element is well placed in its cluster compared

to other clusters. The silhouette values are averaged for all items in the dataset. A good clustering algorithm would produce higher values of the silhouette score.

    We use these two metrics to compare our clustering approach with the hierarchical clustering (in its common implementation using the Ward criterion; Ward (1963) and the k-means clustering algorithm (MacQueen, 1967). Additionally, to determine the effects of the distance metric, we compare the results from our workflow that uses the cosine distance with a version where the

395 pairwise similarity between nodes is computed using the Euclidean distance. Finally, to show the robustness of our approach, we examine various choices for the two free parameters in our workflow, namely the number of reduced dimensions after the PCA (k) and the cluster granularity, which is governed by the disparity filter parameter $\alpha$ used to tune the removal of network edges during the backbone extraction step (Sect. 2.5). Three different values of k are investigated; k=6 corresponding to 50%

of information retained after PCA, k=20 (our choice in the study) corresponding to 72% of retained information, and k=90 corresponding to 95% of retained information. For each value of k, we generate clusters with different $\alpha$ values, with the number of clusters covering 95% of the dataset ranging between 20 and 120.

## 3 Results

### 3.1 The Traits Network

The traits network has 274 nodes representing individual traits and 1422 edges connecting the nodes (Fig. 3). This network encodes the similarity between catchment traits as edge weights. We use the network edges in the clustering to generate groups of traits that are connected to each other, and hence contain similar information. This enables aggregation of our analysis from individual, redundant traits to a smaller number of interpretable, higher-level trait categories that represent similar properties of the catchments.

The size of the clusters range from 35 nodes to 1 node (Table 1), where the 20 largest clusters contain 95% of all nodes in the network. All the clusters are labeled with a unique numerical identifier and are assigned a name that is representative of the trait category. For instance, we refer to cluster number 2 as the "Temperature" trait category since it includes nodes like average annual air temperature for the watershed (T_AVG_BASIN), and other related quantities such as the mean day of the year of first freeze (FST32SITE). Analogously, cluster number 0 includes nodes like the percentages of watershed land covered by development areas at high (DEVHINLCD06), low (DEVLOWNLCD06) and medium intensity (DEVMEDNLCD06). Thus we identify cluster 0 as "Developed Areas" to indicate the presence of these and other traits that represent the extent of human development in the catchment. A full list of traits contained in each cluster of the network is provided in Table F1 in the appendix. A more comprehensive table of trait categories, which includes descriptions of each trait, as well as the traits network topology is provided in the dataset associated with this paper (Ciulla and Varadharajan, 2023) .

| ID | Cluster name | Cluster size | Anthropogenic |
|----|--------------|--------------|---------------|
| 0 | Developed Areas | 35 | yes |
| 1 | Precipitation and Runoff | 31 | no |
| 2 | Temperature | 22 | no |
| 3 | Croplands | 22 | yes |
| 4 | Croplands and Canals | 16 | yes |
| 5 | Croplands and Dams | 15 | yes |
| 6 | Barren Soil and Deciduous Forests | 12 | no |
| 7 | Elevation | 12 | no |
| 8 | Evergreen Forests | 12 | no |
| 9 | Woody Wetlands and Croplands | 11 | yes |

| ID | Cluster name | Cluster size | Anthropogenic |
|---|---|---|---|
| 10 | Lakes, Ponds and Reservoir | 10 | yes |
| 11 | Pastures and Grasslands | 10 | yes |
| 12 | Fine Soils | 10 | no |
| 13 | Major Dams | 9 | yes |
| 14 | Summer Precipitation | 7 | no |
| 15 | Herbaceous Wetlands | 7 | no |
| 16 | Mixed Forests | 6 | no |
| 17 | Coarse Soil | 6 | no |
| 18 | Perennial Ice and Snow | 5 | no |
| 19 | Shrublands | 5 | no |
| 20 | Lower Order Streams | 5 | no |
| 21 | Higher Order Streams | 4 | no |
| 22 | Non Croplands | 1 | yes |
| 23 | Overland Flow | 1 | no |
| 24 | Bulk Density | 1 | no |

Table 1: The list of trait categories identified by our methodology as clusters generated by the network connectivity patterns using the GAGES-II trait dataset. The IDs match the numbers in Fig. 3, while cluster names are humanly assigned to represent the majority of the traits in the cluster. Cluster size refers to the number of nodes in the cluster, and the 'anthropogenic' column indicates if the traits within the cluster are associated with human activities.

## 3.2 The Catchment Network

In the catchments network, each of the 9067 catchments is a node and the 559207 edges connect pairs of catchments with similar traits. The network provides information about catchment similarity, and nodes belonging to a cluster are more similar to each other than the rest of the network. The clusters range from 953 nodes to 1 node in size. The top 71 clusters contain 95% of the nodes in the network, while the 34 clusters with size greater than 50 nodes cover 84% of all nodes. We provide the data for the catchments network topology and partitioning into clusters in Ciulla and Varadharajan (2023).

Figure 4 shows the spatial distribution of catchment clusters using the coordinates of their corresponding gaging stations. We find that catchments within a cluster tend to be located in the same region, even though no geographical coordinates were used as traits in our network methodology. For example, large portions of the Midwestern US are predominantly cluster 2 (green circles) and the coastal area of the Southeastern US are cluster 4 (purple circles). However, cluster 1 (orange circles), is distributed across the CONUS and comprises agglomerates of dense nodes separated by long distances. The global homogene-

ity measure of the catchments is 0.71, which means that the nearest neighbors for 71% of the catchments belong to the same

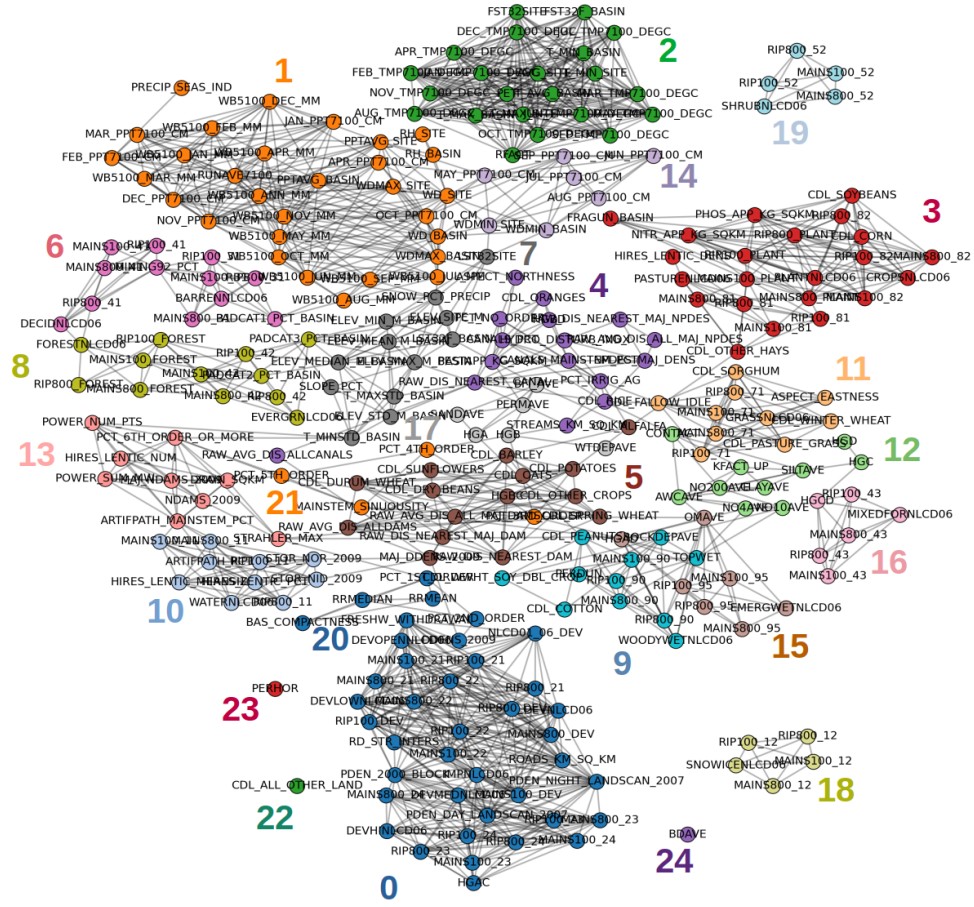

**Figure 3.** The network of traits, where the nodes are traits and edges represent their pairwise similarity. The nodes are colored according to the different clusters and matching color numbers are in descending size order as shown in Table 1.

cluster. This is significantly higher than the homogeneity measure of 0.04 when clusters are randomly distributed throughout the CONUS. In Sect. 3.3, we show how the predominant traits of the catchment clusters can be used to interpret this result.

## 3.3 Characterization of Catchments Clusters with Trait Categories

We infer traits that are over- or under- expressed in each catchment cluster using the z-scores from the traits network. We use two clusters as examples (Fig. 5) to highlight how the network methodology allows interpretation of the characteristics of each catchment cluster. The catchments within cluster 0 (blue circles) are spread across the Western US only, while cluster 1 (orange circles) is distributed across the CONUS. The z-scores of every trait for the two clusters are shown in Fig. 6 (a,b), where each node in the network is sized according to the absolute values of their z-score, while the color reflects the sign (of

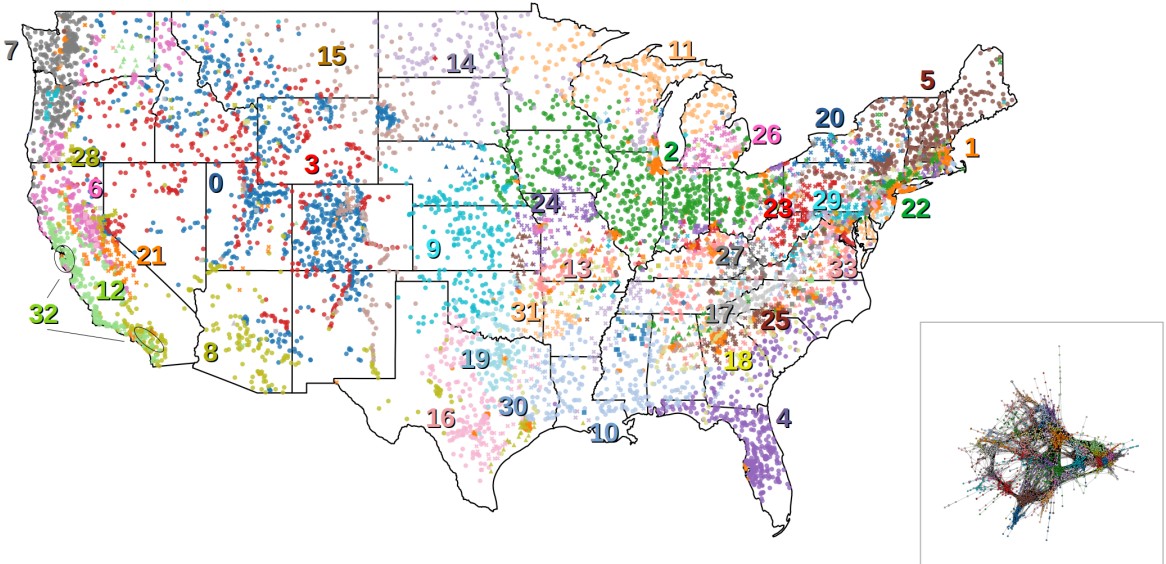

**Figure 4.** Geographical representation of the catchment clusters, where each symbol represents a catchment, and is located at its corresponding gaging station. The catchment network representation with nodes and edges is shown in the figure inset. The numerical identifiers correspond to the IDs in Table 2. The colors depict different clusters, and are consistent between the map and the network diagram. Note that due to the lack of a palette with enough distinguishable colors, some of the clusters are indicated using different symbols when colors overlap.

over or under expression). For both catchment clusters, the respective traits network contains entire groups of over and under
expressed nodes.

Leveraging the trait clusters, we condense the information of the unique expression of the 274 traits into 25 interpretable
categories by computing the cluster-wide average of the traits z-scores (see examples in Fig. 6 c,d). Cluster 0 (blue circles)
shows over expression of the category Elevation (z-score=1.49) and Evergreen forests (z-score=1.04) and under expression
of the Temperature category (z-score=-1.28). Based on the expression of these and other trait categories, we infer that the
blue cluster generally represents mountainous forested areas. Cluster 1 (orange circles) is dominated by the trait category
named Developed areas (z-score=2.29), indicating the catchments represented by the orange nodes are located in proximity
to developed areas such as cities. The fact that this cluster is dominated by an anthropogenic category is consistent with the
heterogeneous nature of its spatial pattern. In contrast, the rather homogeneous distribution of catchment cluster 0 is likely due
to the prominence of climatic trait categories that are more correlated with the geographical location.

The list of trait categories and their relative z-score values associated with all the catchment clusters identified using our
methodology is provided in the dataset Ciulla and Varadharajan (2023). A brief summary for clusters with size greater than 50
is shown in Table 2 along with their corresponding spatial homogeneity measures.

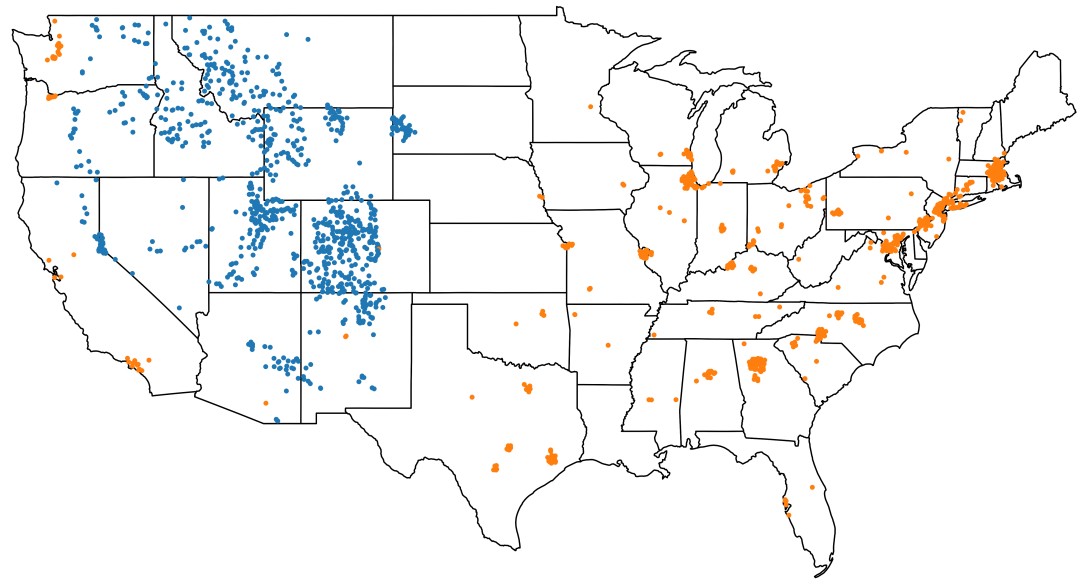

**Figure 5.** Spatial distribution of two of the catchment clusters from Fig. 4, shown separately for easier visualization. Maps showing the spatial distributions individually for the 34 clusters of size greater than 50 are included in the file S2 of the dataset associated with this paper (Ciulla and Varadharajan, 2023)

| ID | Size | Brief Descriptor of Dominant Trait Categories | Homogeneity |
|---|---|---|---|
| 0 | 953 | Low Temperature, High Elevation, Evergreen Forests | 0.53 |
| 1 | 729 | Developed Areas | 0.12 |
| 2 | 673 | Croplands, Fine Soils | 0.57 |
| 3 | 450 | High Elevation, Low Summer Precipitation, Shrublands | 0.28 |
| 4 | 419 | High Temperature, Low Elevation, Wetlands | 0.79 |
| 5 | 395 | Low Temperature, High Summer Precipitation, Mixed Forests | 0.49 |
| 6 | 341 | High Precipitation and Runoff, High Elevation, Low Summer Precipitation, Evergreen Forests | 0.33 |
| 7 | 337 | High Precipitation and Runoff, Evergreen Forests | 0.80 |
| 8 | 292 | Low overall Precipitation and Runoff, High Temperature, Shrublands | 0.28 |
| 9 | 240 | Pastures and Grasslands | 0.69 |
| 10 | 214 | High Temperature, Low Elevation, Woody Wetlands and Croplands | 0.55 |
| 11 | 210 | Low Temperature, Lakes and Reservoirs, Wetlands | 0.63 |
| 12 | 202 | Low Summer Precipitation, Mixed Forest, Shrublands | 0.70 |
| 13 | 195 | High Summer Precipitation, High Temperature, Croplands | 0.17 |

| 14 | 186 | Low Temperature, Herbaceous Wetlands, Croplands | 0.53 |
|----|-----|------------------------------------------------|------|
| 15 | 177 | Low Precipitation, Pastures and Grasslands | 0.19 |
| 16 | 170 | High Temperature, Shrublands | 0.67 |
| 17 | 138 | High Precipitation and Runoff, Barren Soil and Deciduous Forests | 0.82 |
| 18 | 132 | High Temperature, Major Dams | 0.07 |
| 19 | 124 | High Temperature, Pastures and Grasslands | 0.71 |
| 20 | 112 | Low Temperature, High Summer Precipitation, Croplands, Mixed Forests | 0.36 |
| 21 | 108 | High Elevation, Barren Soils and Deciduous Forests | 0.10 |
| 22 | 100 | High Summer Precipitation, Lakes and Reservoirs | 0.11 |
| 23 | 98 | High Summer Precipitation, Barren Soils and Deciduous Forests | 0.54 |
| 24 | 95 | Croplands, Fine Soils | 0.63 |
| 25 | 76 | High Temperature, Low Elevation, High Summer Precipitation | 0.30 |
| 26 | 69 | Wetlands and Croplands | 0.59 |
| 27 | 64 | High Summer Precipitation, Barren Soils and Deciduous Forests | 0.76 |
| 28 | 63 | Low Temperature, High Elevation, Low Summer Precipitation, Lakes and Reservoirs | 0.07 |
| 29 | 61 | High Summer Precipitation, Croplands | 0.62 |
| 30 | 61 | High Temperature, Major Dams, Woody Wetlands and Croplands | 0.17 |
| 31 | 54 | High Temperature, High Summer Precipitation, Mixed Forests | 0.29 |
| 32 | 53 | High Temperature, Low Summer Precipitation, Developed Areas | 0.19 |
| 33 | 51 | High Temperature, Low Elevation | 0.12 |

Table 2: Table of dominant trait categories and spatial homogeneity measures of the 34 catchment clusters with at least 50 nodes. The clusters are sorted by decreasing size (i.e., number of nodes in the cluster). Values shown are the cluster unique ID, cluster size, a brief descriptor based on the dominant over and under expressed traits categories, and cluster geographical homogeneity measure.

## 3.4 The Representative Catchments

Here, we provide an example of how our methodology can be used to identify representative catchments using the two clusters highlighted in Sect. 3.3. A total of 10 catchments are considered representative of cluster 0, and 14 are considered representative of cluster 1 (Fig. 7). In both cases, the selected catchments represent at least 95% of the nodes in each cluster. The selected representative nodes are geographically distributed within the spatial domain of the cluster they belong to. This behavior is the result of the combined effect of (1) the intrinsic multiscale connection patterns in the network, where heterogeneity at different scales produces groups of more densely connected nodes within the clusters, and (2) the strategy of node selection,

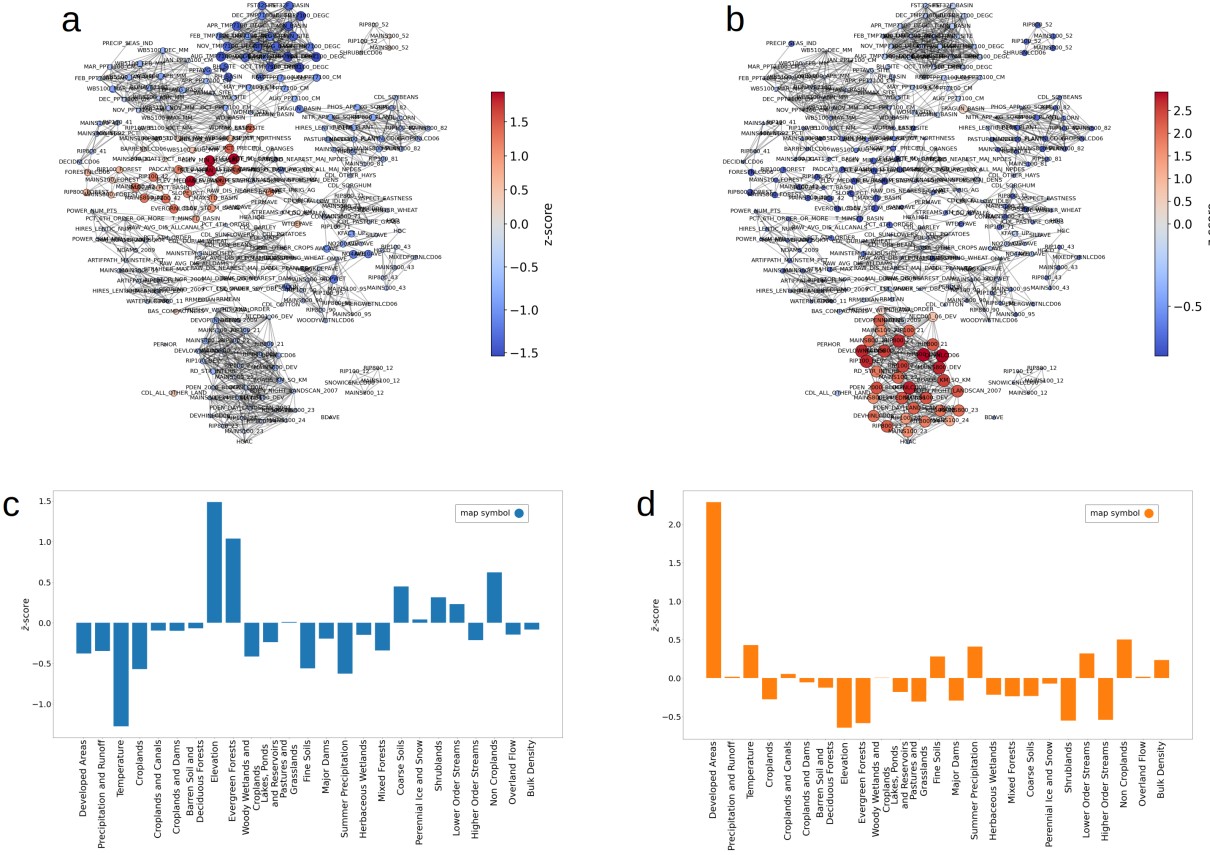

**Figure 6.** Network of traits for (a) cluster 0 (blue circles) and (b) cluster 1 (orange circles) shown in Fig. 5. Nodes are colored according to the z-score values computed for the two catchment clusters. Node size is proportional to the absolute values of the z-score. Traits that are over expressed are shown in red, and those that are under expressed are shown in blue. In this way nodes depicting over and under expressed traits have bigger nodes and are more visible. The cluster aggregated z-scores are shown as barcharts for (c) the cluster 0 (blue circles) and (d) cluster 1 (orange circles) from Fig. 5.

which promotes representative nodes to be spread across each cluster. These representative catchments can be used to prioritize locations for observations or modeling purposes.

### 3.5 Hydrological Indices and Trait Categories

The results of the statistical tests are summarized in appendix in Table G1 for all 34 streamflow indices. When averaged for all the indices, 83% of the clusters for the 1-sample K-S test and 79% for the 2-sample K-S test reject the null hypothesis, meaning that the distribution of their streamflow indices is mostly distinct from the overall distribution at the continental scale.

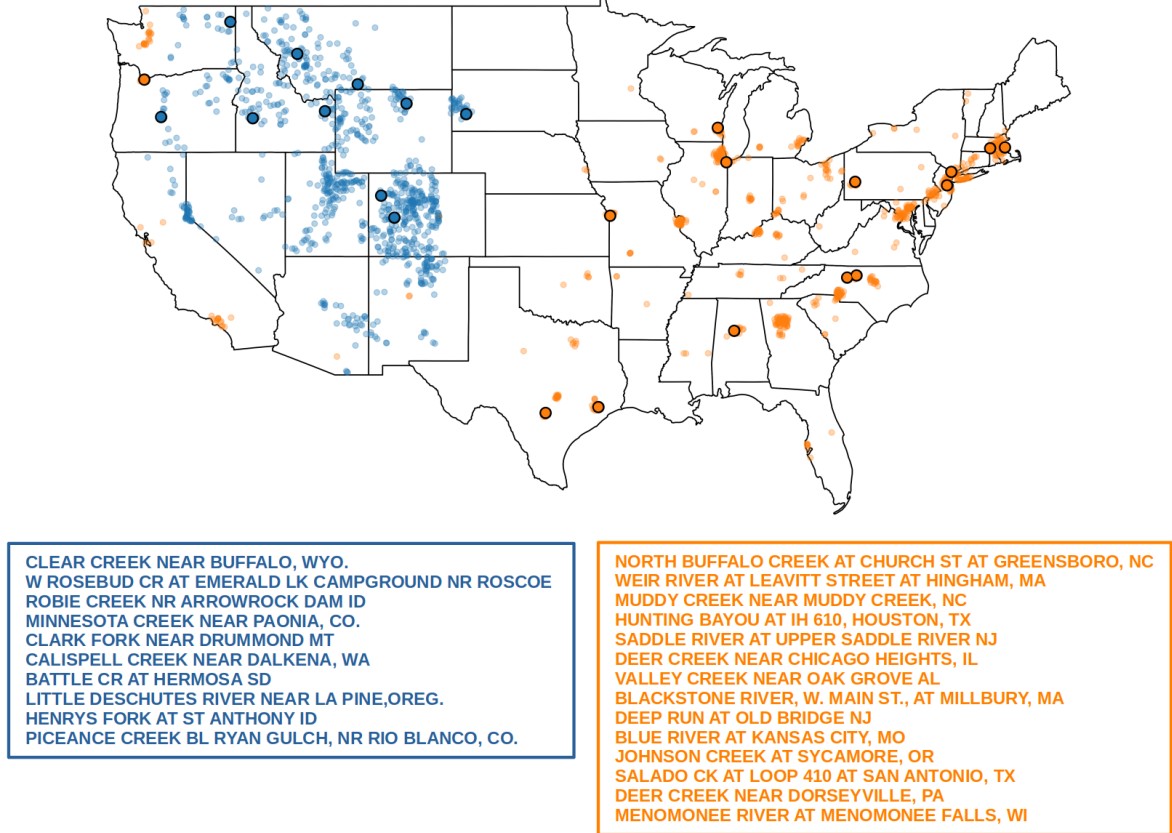

**CLEAR CREEK NEAR BUFFALO, WYO.**
**W ROSEBUD CR AT EMERALD LK CAMPGROUND NR ROSCOE**
**ROBIE CREEK NR ARROWROCK DAM ID**
**MINNESOTA CREEK NEAR PAONIA, CO.**
**CLARK FORK NEAR DRUMMOND MT**
**CALISPELL CREEK NEAR DALKENA, WA**
**BATTLE CR AT HERMOSA SD**
**LITTLE DESCHUTES RIVER NEAR LA PINE,OREG.**
**HENRYS FORK AT ST ANTHONY ID**
**PICEANCE CREEK BL RYAN GULCH, NR RIO BLANCO, CO.**

**NORTH BUFFALO CREEK AT CHURCH ST AT GREENSBORO, NC**
**WEIR RIVER AT LEAVITT STREET AT HINGHAM, MA**
**MUDDY CREEK NEAR MUDDY CREEK, NC**
**HUNTING BAYOU AT IH 610, HOUSTON, TX**
**SADDLE RIVER AT UPPER SADDLE RIVER NJ**
**DEER CREEK NEAR CHICAGO HEIGHTS, IL**
**VALLEY CREEK NEAR OAK GROVE AL**
**BLACKSTONE RIVER, W. MAIN ST., AT MILLBURY, MA**
**DEEP RUN AT OLD BRIDGE NJ**
**BLUE RIVER AT KANSAS CITY, MO**
**JOHNSON CREEK AT SYCAMORE, OR**
**SALADO CK AT LOOP 410 AT SAN ANTONIO, TX**
**DEER CREEK NEAR DORSEYVILLE, PA**
**MENOMONEE RIVER AT MENOMONEE FALLS, WI**

**Figure 7.** Geographical representation of clusters 0 and 1, highlighting locations of their representative nodes. The catchment names, as specified in the GAGES-II dataset, are shown in the figure legend with same color as the circles in the map.

These results show that the trait-based clustering approach results in distinct signature classification. See Sect. 4.5 for further discussion about the distinct hydrological behavior across the catchment clusters.

### 3.6 Comparison with Traditional Clustering Techniques

The two metrics, cluster similarity and silhouette score, indicate that our workflow performs better than traditional unsupervised methods of classification (Fig. 8). We also find that the network-based clustering (red and green points) is considerably superior to both k-means (yellow points) and hierarchical clustering (blue points) across the different values of k and cluster granularity (Fig. H1 in appendix). This is evident from the consistently low values of the median cluster similarity and higher values of silhouette scores for our methodology. Also, the network generated using the cosine distance as a similarity metric (red points) performs better than its counterpart that uses the Euclidean distance (green points). This confirms that the cosine similarity should be preferred as a distance metric in high dimensions and the directionality of the data can contain valuable information.

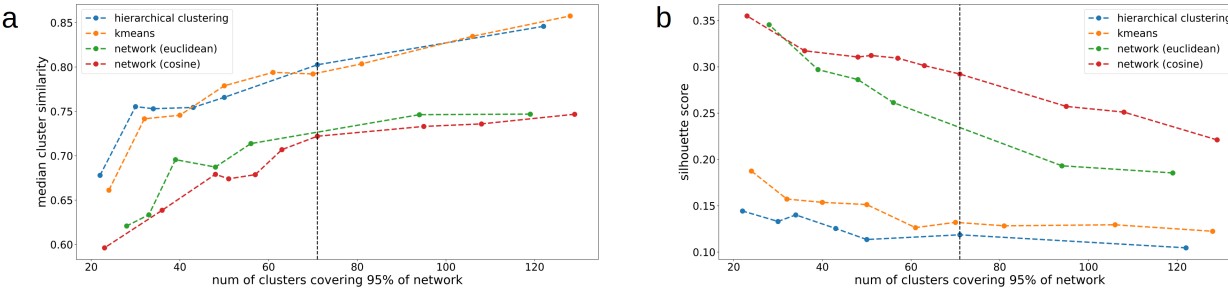

**Figure 8.** Median cluster similarity values (a) and silhouette scores (b) for different clustering methods and similarity measures used in the network analysis. The number of reduced dimensions after PCA is equal to 20, corresponding to 72% of retained information. The vertical black dashed line refers to the cluster granularity used in the paper. The colored dashed lines are shown for visualization of trends. Lower values of the median cluster similarity metric (a) correspond to better clustering performance. Higher values of the silhouette scores (b) correspond to better clustering performance.

## 4    Discussion

### 4.1    Advantages of our network-based workflow as a tool for classification

In this study, we present a novel methodology to classify river catchments in an unsupervised manner using network science. We chose to use the GAGES-II dataset as the source of information for catchment characterization because of its comprehensive set of over 300 natural and anthropogenic traits across more than 9000 catchments in the CONUS. In our methodology, we use the combination of the PCA algorithm, networks, and the cosine similarity metric to mitigate the issues of information redundancy and high dimensionality in the data. In particular, the distance metrics perform worse (referred to as "degradation") as the number of dimensions grows (Aggarwal et al., 2001). One of the consequences of this phenomenon is that the ratio of the distances of the nearest and farthest neighbors to a given point in high dimensions approaches 1, meaning that the points become uniformly distant from each other (Beyer et al., 1999). The PCA partly resolves these issues by projecting the original high dimensional vector data into a smaller set of orthogonal directions called principal components. However, there are limitations to how small the resulting vectors can be, because data compression inevitably results in some loss of information. It is not uncommon to retain a high number of dimensions after PCA (20 in our case), which still results in degradation of the distance metric and has some redundant information. Thus applying K-means or hierarchical clustering algorithms after PCA on large dimensional datasets is not a solution to addressing the issues of multicollinearity and high data dimensionality, given the dependence of these methods on the Euclidean distance.

Hence, we use networks to model the relationships amongst elements of the system represented by the high dimensional vectors. The elements are the nodes of a network connected by edges that represent their pairwise distance metric, which are not bound to be points in a vectorial space. Instead we use their vectorial representation to compute any appropriate distance metric and that information is reflected in the connection between nodes as edge weights. Thus, the task of finding clusters in

the Euclidean space is shifted to the identification of connectivity patterns in a network. In this respect, the network constitutes a generalized tool for system analysis where edges can represent different types of distance metrics or other pairwise relational quantities. In our study, we chose to use the cosine distance to describe the relationships amongst the reduced vectors of both traits and catchments since it is more appropriate for use in high dimensions and preserves the directionality of the data. The rationale for the choice of this similarity metric over the Euclidean distance is discussed in Appendix C. The use of the network enables us to move to this alternate distance metric, surmounting issues encountered by common methods like the K-means and Ward's hierarchical clustering that rely on the Euclidean distance to generate the division into classes. Once we set up our investigation as a network analysis the problem of classification is translated into finding clusters in the networks. Many clustering algorithms are available for network science (Fortunato, 2010), and our method implements the widely used Infomap algorithm (Rosvall and Bergstrom, 2008).

We have demonstrated that our network-based workflow outperforms the K-means or the Ward's hierarchical algorithms for two different metrics (see Sect. 3.6), particularly with the parameters we chose for dimensionality reduction and the disparity filter (black dashed lines in Fig. 8 corresponding to $k = 20$ dimensions retained). We conclude that our method produces more distinct clusters, and is hence a better choice than traditionally used classification approaches.

In addition to addressing the challenges of information redundancy and data dimensionality, our analysis using the catchment-trait networks is also interpretable, as it can quantitatively determine trait categories that are over or under expressed in catchment clusters (Sect. 4.2). Additionally, we are able to use a variety of network-specific metrics that highlight the role of the elements of the system at different scales of the network. For example, we use the degree centrality computed using the nearest neighbors of a node as a local measure to identify catchments that have a high number of connections to others. In a similarity network, a node with a high degree measure means it is similar to many other nodes, and hence we use the degree centrality to generate sets of representative catchments. At an intermediate scale, the clustering coefficient reveals how many neighbors of a node are connected to each other, revealing heterogeneous, non-trivial connectivity patterns that indicate the presence of clusters. At a global scale across the network, we leverage the flow of information traveling through the nodes in the Infomap algorithm (Rosvall et al., 2009) to generate the clusters. Additionally, statistical methods can be applied to select the information contained in the edges of the network (see Sect. 4.4.2 for an example). Although we demonstrate the use of this workflow for catchment classification, the methodology can easily be extended to analyze other types of environmental trait datasets such as microbial or plant traits.

## 4.2 Interpretability and Redundancy Analysis

Machine learning techniques are capable of proficiently learning patterns in the data (Bishop and Nasrabadi, 2006), but are often black-box methods that are difficult to interpret. In the case of supervised approaches, the accuracy of a method can be evaluated using a test set, which is not an option for unsupervised algorithms because of the lack of labeled data. Hence, we simultaneously analyze the information from the networks of traits and catchments to interpret the results for a deeper understanding of the specific set of traits that result in classification of a catchment cluster.

Specifically, we characterize the clusters of catchments by investigating the traits that are over or under expressed, which are grouped into a small, interpretable set of trait categories. The generation of the higher-level trait categories reduces the number of redundant traits by aggregating them into clusters, which allows for easier human interpretation. For example, a set of traits related to air temperatures, when clustered together are no longer isolated measures and are easier to interpret when presented as an aggregated "Temperature" category. To illustrate how our approach helps with reducing trait redundancy, we computed the spearman correlation coefficients ($\rho$) between streamflow indices and catchment traits, which account for non linearities in the data. We find that traits belonging to the same trait categories have similar correlations with the streamflow indices (Fig. I1 in appendix). We find that the spearman correlation coefficients between the streamflow indices and individual catchment traits can be effectively represented as an aggregated median value for the trait categories generated in our method (Fig. 10), which indicates that our reduced set of trait categories are sufficient to determine the general relationships between traits and hydrological signatures. Thus, interpretability is enabled with the generation of a small number of trait categories whose composition can be easily visualized and statistically analyzed.

Topological proximity at the cluster level is also meaningful. Clusters that share a connection, and are close to each other, are also more related to each other. For example, cluster 3 in the trait network, which includes agricultural traits, and cluster 5, which has traits related to cropland as well as dams are topologically close with many shared edges, and also appear intuitively to be related to each other.

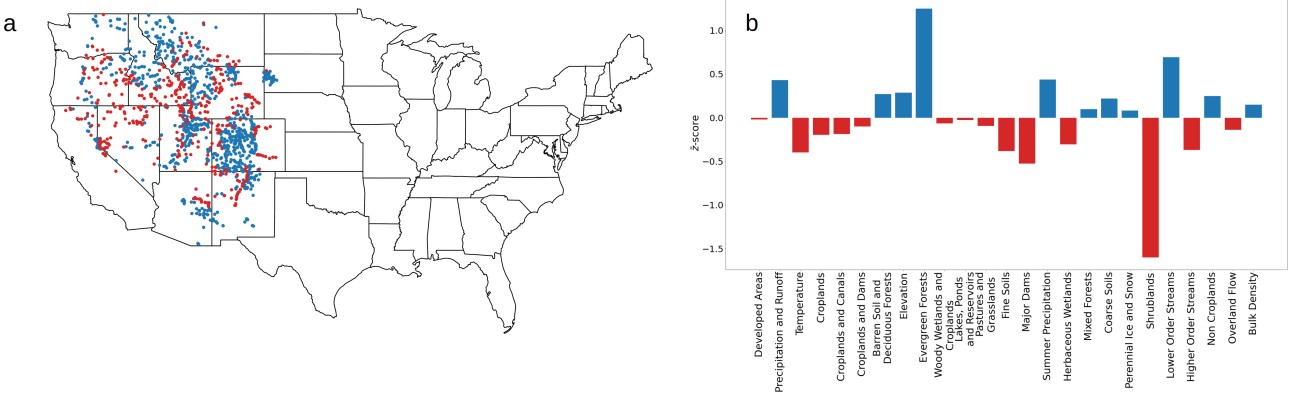

**Figure 9.** (a) Geographical representation of the two catchment clusters that are spread across most of the Western US. (b) Bar charts showing the difference in z-scores of the trait categories between the two clusters shown in the map. Positive values denote categories more present in cluster 0 (blue circles), while negative values are more present in cluster 3 (red circles).

Finally, the identification of dominant traits is not only useful to characterize individual catchment clusters (Fig. 6), but can also be used to compare across different clusters. For example, in Fig. 9 we show the differences between the traits categories of two catchments clusters located in the Western US. Since the cluster locations overlap geographically, we use the differences in the z-score values of the trait categories to understand how the groups were split by identifying which categories are relatively

over expressed in the two clusters. We can deduce that cluster 0 refers to catchments with higher precipitation and runoff, higher elevation and the presence of evergreen forests, while the cluster 3 has higher temperatures and shrublands. Note that these are relative quantities, and in fact cluster 0 overall has lower precipitation and runoff in comparison to the rest of the network as shown in Fig. 6. In the rest of the discussion sections, we leverage the trait categories associated with catchment clusters to reveal insights relevant to their hydrological behavior.

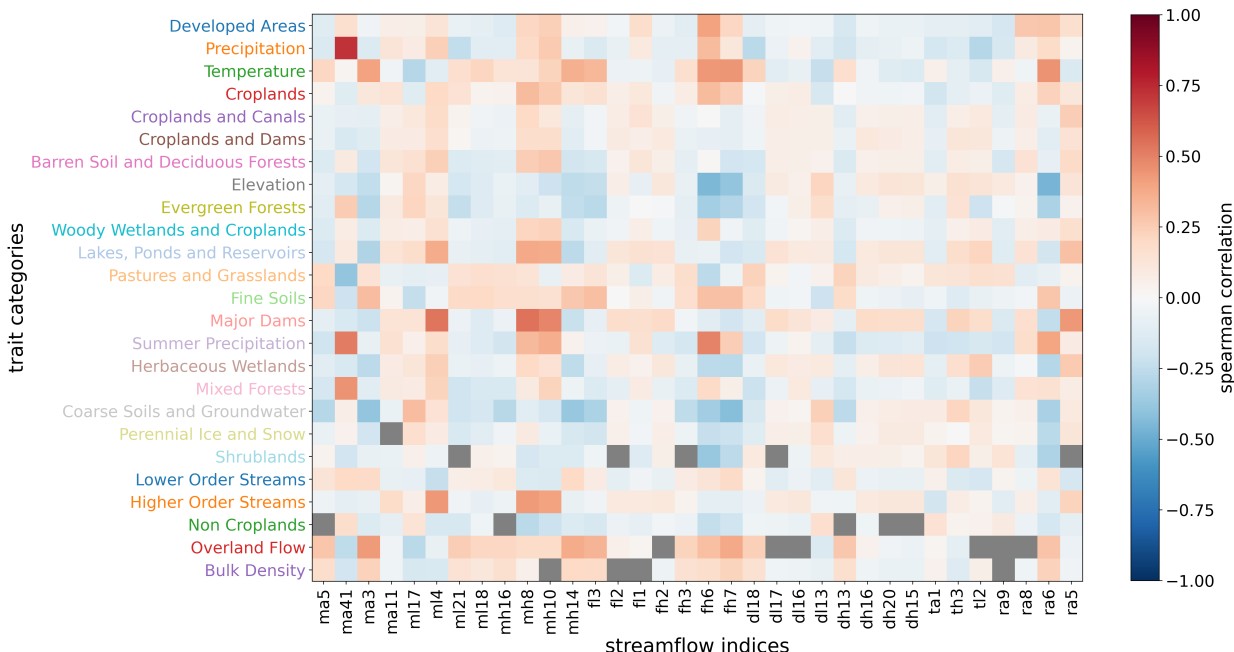

**Figure 10.** Heatmap showing spearman correlation coefficients between streamflow indices and the traits categories generated in the study. The intensity of the colors show the degree of correlation or anticorrelation as indicated in the color bar. Gray boxes indicate correlations that are above the significance level of p>0.05.

## 4.3 Spatial Homogeneity of Catchment Clusters with Natural and Anthropogenic Traits

The results in Sect. 3.5 show that in general the catchment clusters tend to be geographically co-located throughout the CONUS, with a few exceptions. This means that the trait-based classification results in clusters that have some geographical homogeneity, which would not occur with a random distribution of clusters. However, at the cluster level, we see a high variability of spatial homogeneity as shown in Table 2. This reveals that the geographical domain generated by each cluster (i.e., its convex hull), contains catchments from other clusters, and the extent of the relative abundance changes across clusters.

We leverage the information from the trait categories to investigate this variability. We find that most of the trait clusters are composed of mostly anthropogenic or natural traits only. This result not only provides additional validation of the trait

classification, but it can be used to identify the effects of human influence on the hydrological behavior of catchment clusters. Thus we divide the catchment clusters into three groups of increasing (low, mid, high) human influence. Our hypothesis is that clusters dominated by natural factors, with little human disturbance, should exhibit more spatial homogeneity as a consequence

of the environmental forcing. Conversely, clusters with strong anthropogenic influences, like urban areas or dams, are less coupled to natural factors and should display lower homogeneity. A middle ground is constituted by clusters where the human impact is interlaced with natural factors like in croplands. We first sorted the catchment clusters from the lowest to the highest value of cluster homogeneity (Fig. 11a). The six clusters with strong anthropogenic influence are amongst the last ten with respect to their spatial homogeneity, confirming that the human activities are associated with a more heterogeneous spatial

distribution of catchment clusters. However, none of the clusters have a homogeneity measure lower than the one assuming a random cluster distribution. Note that the spatial homogeneity is computed using geographical coordinates and does not account for the elevation of the catchments. This could explain lower homogeneity values in some of the catchment clusters dominated by natural factors.

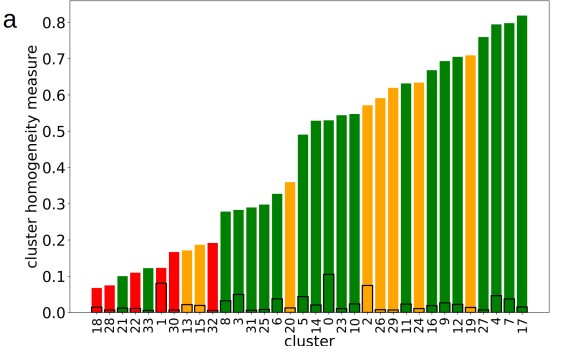 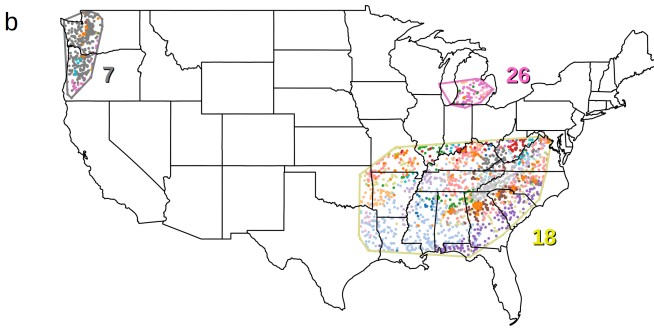

**Figure 11.** (a) Bar chart of the cluster homogeneity measures sorted in ascending order. The colors represent the anthropogenic influence, with high values in red, low in green and mid (i.e. interlaced human and natural factors) in yellow. The bars shown as black lines represent values assuming random co-location of clusters. (b) Examples of convex hulls for a cluster with significant human influence (cluster 18), one with predominantly natural traits (cluster 7), and a cropland dominated cluster (cluster 26). The color of the convex hull polygon matches the color of the catchments cluster that generated it.

## 4.4 Trait Relevance at Different Spatial Scales

So far, our study has only focused on the classification of the catchments at the continental scale. However, our proposed method can be applied at multiple spatial scales as long as relevant trait information is available. Here, we show two different approaches that can be used for catchment classification at different scales.

### 4.4.1 Multiscale analysis at regional and continental scales

First, we apply the workflow previously used for the CONUS scale at the regional scales to obtain the trait and catchment
networks. We demonstrate this approach for the Upper Colorado River Basin (UCRB; USGS HUC 14). There are 368 catchments within this area, which is about 25 times smaller than the total number across the CONUS. Similar to the results from the CONUS-scale, we find a well-defined geographical coherence among the nodes of each catchment cluster (Fig. 12).

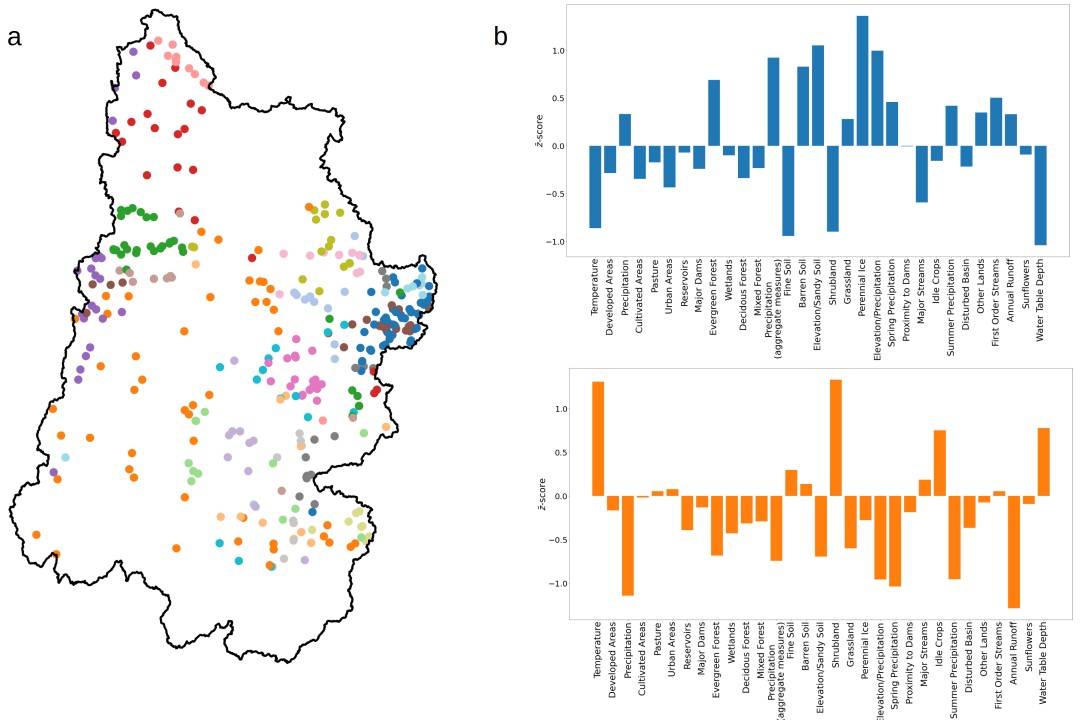

**Figure 12.** (a) Geographical representation of the network of catchments in the Upper Colorado River Basin, with the HUC 14 boundary shown in black and the locations of the gaging stations used to map the catchment clusters (similar to Fig. 4). (b) Bar charts of the z-scores of the trait categories for two of the clusters in this basin. The colors in the map and the bar charts match. The blue nodes are located on the east side of the basin while the orange ones are distributed in the lower half of the basin.

Not surprisingly, the trait network connectivity and cluster composition changes to reflect the characteristics of this subset of catchments. The characteristic z-scores calculated for this subset can provide interpretation of the traits expression in this
region. New trait clusters emerge that are more relevant to the UCRB, like the one representing areas at high elevation covered with sandy soils. Some trait clusters, like the one consisting of temperature-related traits or the cluster associated with shrubland-dominated environments are present at both the CONUS and this particular regional scale. The complete list of traits network clusters for the UCRB is available in Ciulla and Varadharajan (2023).

### 4.4.2 Edge Weights Filter Tuning

Another way to capture insights at different spatial scales is to tune the edge filtering parameter as discussed in Sect. 2.5. By changing the significance level of the disparity filter, we tune the connectivity of the network of catchments to generate larger or smaller clusters. There is no prescribed granularity at which the system should be investigated, and we can make adjustments to fit the goals of the analysis. Here, we provide an example where a smaller value of the significance level acts as a higher threshold for the edge weights, reducing the number of edges in the network of catchments and producing smaller clusters. Here, cluster 4, generated using the original filter threshold that was spread across most of the South Atlantic US (purple circles in Fig. 4) is split into the two new clusters as shown in Fig. 13. We again use the difference between the z-scores of the trait categories in the two newly generated clusters to interpret the partitioning. We find that the difference is primarily due to separation of catchments that have predominant traits as Herbaceous Wetlands, Croplands and Canals and moderately higher temperatures and summer precipitation, which tend to be concentrated in the Florida peninsula, in contrast to those with Evergreen Forests and Lower Order Streams, which were more prevalent in the rest of the original cluster (Fig. 13b).

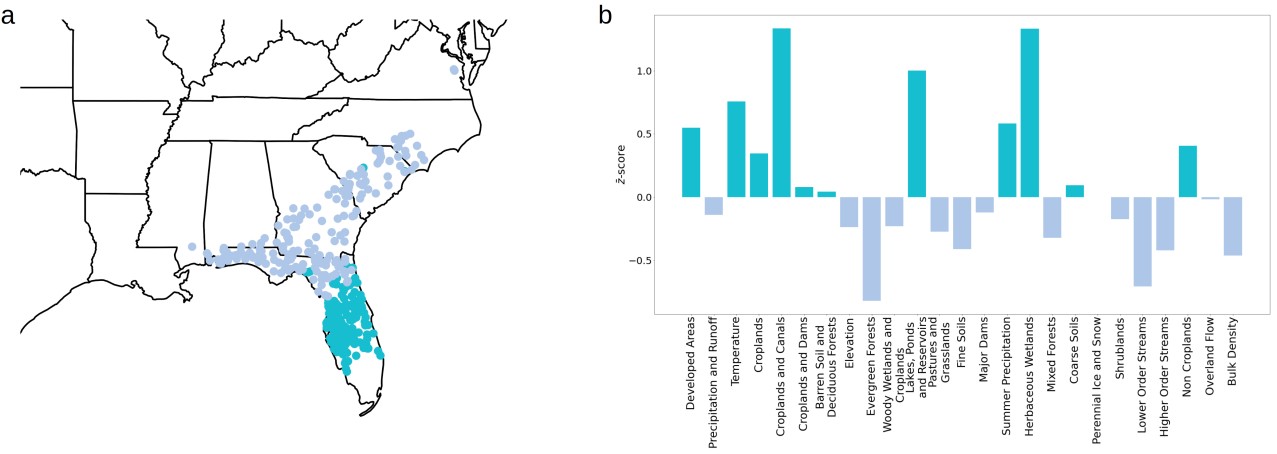

**Figure 13.** (a) Geographical representation of two clusters, shown in different colors, of the network of catchments generated with smaller significance level of the disparity filter. The union of the nodes of these two clusters corresponds to the cluster 4 in the South Atlantic US in the original catchment network that are shown as purple circles in Fig. 4. (b) Bar chart of the difference between the z-scores of the two clusters. The color of the bars matches the cluster where the category is over expressed.

## 4.5 Distinct Hydrological Behavior of Catchment Clusters

In this study, we have so far analyzed the characteristics of river catchments at regional to CONUS scales with their physical traits. Here we extend our analysis to examine possible connections between the catchment clusters and their hydrological behavior. In Sect. 3.5, we show that the distributions of hydrological indices calculated using stream flows are statistically

different when aggregated by catchment cluster, indicating that the partitioning with the traits generates clusters that have distinct hydrological behavior.

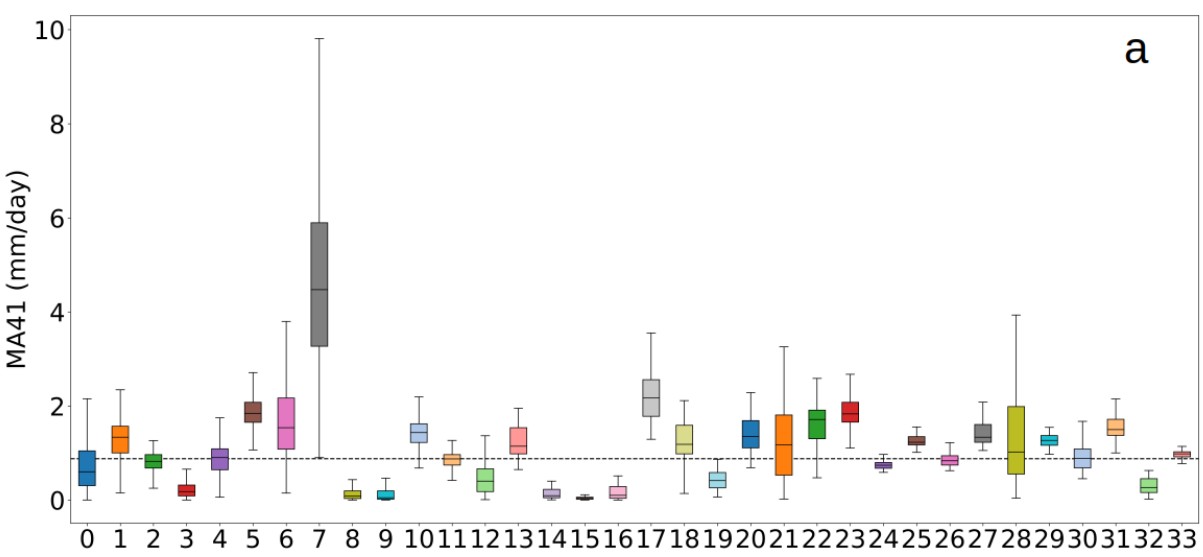

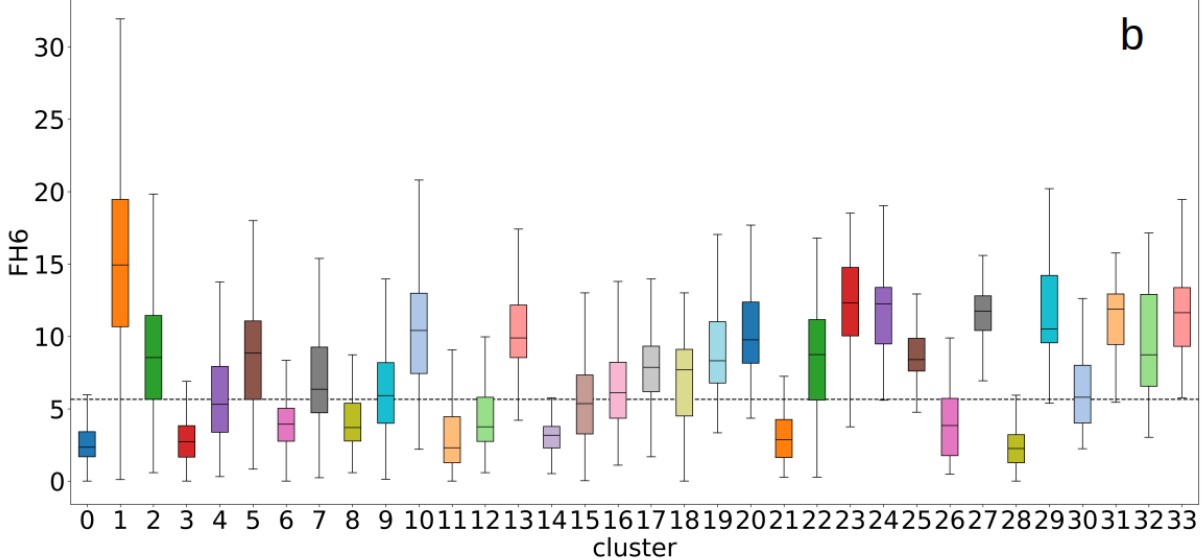

**Figure 14.** Boxplot of the hydrologic indices ma41 (mm/d) (a) and fh6 (b) representing the mean annual runoff and the yearly average number of moderate floods events per catchment respectively. The x-axis shows the catchment clusters of size greater than 50 sorted in descending size order. The values in the y axis are indices aggregated by catchment clusters. The colors of the bars match the colors of the clusters in Fig. 4.

We can leverage the interpretability provided by the trait categories to hypothesize predominant factors that influence the distinct hydrological behavior in the catchment clusters. To illustrate this, we first examine the distributions of two indices from Olden and Poff (2003), mean annual runoff (ma41) and the average yearly number of moderate floods events in a watershed (fh6) (see Fig. 14). In fh6, a flood is considered moderate when the stream flow exceeds 3 times the value of the median flow. Cluster 7 (gray box in Fig. 14) displays the highest median and the highest upper whisker for ma41, which is the 75th percentile plus 1.5 times the interquartile range. The catchments in this cluster are predominantly located in the Pacific Northwest (Fig. 4). From the analysis of the over expressed trait categories, we find that this cluster is characterized by high precipitation. Not surprisingly, we find catchments in the Pacific Northwest with higher precipitation also have greater average flows. Cluster 1 (orange box in Fig. 14b) displays the highest median and the highest upper whisker for fh6, and contains catchments in proximity to developed areas (Fig. 6d). Hence we can hypothesize that predominantly developed catchments are more vulnerable to moderate flooding, which is consistent with a previous finding from Ombadi and Varadharajan (2022) showing that catchments with a higher percentage of impervious cover in the GAGES-II dataset generally have increased runoff ratios in both temperate and arid climates. The boxplots of the full set of hydrological indices included in this study are presented in Ciulla and Varadharajan (2023).

Additionally, we utilize the ability to determine correlations between stream flow indices and a reduced set of interpretable trait categories (Sect. 4.2). We find that across the 9067 catchments, mean annual runoff (ma41) is not just positively correlated with traits related to precipitation, but also with the presence of mixed forests ($\rho = 0.45$) and to a lesser degree evergreen forests ($\rho = 0.25$). The ma41 index is also negatively correlated with the 'pastures and grasslands' trait category ($\rho = -.40$). This highlights the role of vegetation in mediating flows, and is somewhat counterintuitive given that in the absence of management, forested catchments with higher evapotranspiration would be expected to have lower flows compared to grasslands. Similarly, the fh6 index is not just positively correlated with precipitation traits, but also with traits related to developed areas ($\rho = 0.40$), croplands ($\rho = 0.32$) and temperature ($\rho = 0.43$). It is also inversely correlated with elevation ($\rho = -0.45$), presence of shrublands ($\rho = -0.38$), evergreen forests ($\rho = -0.28$), coarse soils and groundwater ($\rho = -0.35$). These relationships are consistent across other flood indices. For example, the fh7 index showing the propensity for heavy floods (above 7 times median flows) similarly has a moderate positive correlation with temperature ($\rho = 0.44$) and overland flow ($\rho = 0.38$), and a moderate negative correlation with elevation ($\rho = -0.39$) and coarse soils/groundwater ($\rho = -0.43$). This indicates how flooding is affected by the complex relationships between land use, vegetation, soil infiltration capacity and base flows.

These examples highlight the use of our methodology to demonstrate how specific hydrological behaviors can be connected to catchment traits such as their climatic conditions, topography, land use or anthropogenic influence. The ability to link catchment traits to specific hydrological behaviors, enables further analysis of the factors that influence different stream flow characteristics (e.g. high versus low flows). In particular, the distinction between anthropogenically-influenced trait categories and natural traits (see Table 1) enables further analysis of human activities on hydrologic behavior.

## 4.6 Examining Diversity of Hydrologic Behavior within Catchment Clusters

We can also use our methodology to gain insights into the traits that may result in diversity of hydrologic behavior within similar catchments. For this purpose we selected catchment subsets that are considered outliers - i.e. where the index is either above the 90th percentile or below the 10th percentile of all indices in the cluster, for each streamflow index and for each cluster of catchments. We compare the z-scores (Z) of the traits associated with the catchment subsets relative to the entire catchment cluster to evaluate whether there are differences in traits that would explain the anomalous hydrological behavior.

As an example, we look at catchments within a cluster that have distinct baseflow regimes, based on a baseflow index (ml17) in Olden and Poff (2003) that represents the 7-day minimum flows divided by mean annual daily flows. The results for anomalous catchments have higher than normal (>90th percentile) baseflow are shown in Fig. 15, where the size and color of the bubbles are the relative z-scores of the trait categories.

We focus on a crop-dominated catchment cluster such as the one generally encompassing the Ohio Valley region (cluster 2),
displayed in the third row of the bubble plot. This cluster is characterized by relatively low elevation, presence of croplands and fine soil as indicated by the higher z-scores of these trait categories relative to the rest of the CONUS catchments (Fig. 16a). Using our approach, we can identify the over and under expressed traits of the catchments with anomalously high baseflows in cluster 2 that generally has low elevation croplands. In Fig. 15, we find there is a positive association of high baseflows with coarse soils (Z=0.98) and a negative one with fine soils (Z=-0.51), which is not surprising. In addition, there is an association
of high baseflows with the "Non-cropland" trait category (third last column with green label, Z=0.85), which aggregates all non-agricultural land use such as urban areas and forests. This indicates that within the context of a cropland-dominated cluster, the catchments that have relatively lower areas of croplands have higher baseflows. Interestingly, there is also a strong positive association of high baseflows with shrubland (Z=1.12) and a moderate negative association with temperature (Z=-0.65). One possible explanation for these results is that pumping groundwater for agriculture decreases the groundwater input into streams
resulting in lower baseflows. This depletion of groundwater discharge into streams does not occur in shrublands or other areas without croplands.

Another catchment cluster with a strong agricultural presence is cluster 14, generally located in North and South Dakota, which are characterized by low temperatures, herbaceous wetlands and croplands (Fig. 16b). Similar to cluster 2, there is a positive association of anomalously high baseflows with coarse soils (Z=0.76) and non-croplands (Z=1.10), and a negative
association with fine soils (Z=-0.84). However, in comparison to cluster 2, several other factors have a positive association with high baseflows including precipitation/summer precipitation (Z=0.90, Z=1.04 respectively), the presence of lakes, ponds and reservoirs (Z=1.19), herbaceous wetland areas (Z=0.76), evergreen/mixed/deciduous forests (Z=0.58, Z=0.74, Z=0.88 respectively), and developed areas (Z=0.66). There is also a negative association with overland flows (Z=-0.82). This reveals that, in catchment cluster 14, anomalously high baseflows are more likely in the presence of surface water bodies such as lakes
and wetlands that have the potential for increased surface-groundwater exchange. High baseflows also occur in forested areas of these agricultural catchments, potentially indicating that the partitioning of precipitation is weighted towards infiltration and recharge over evapotranspiration in these catchments.

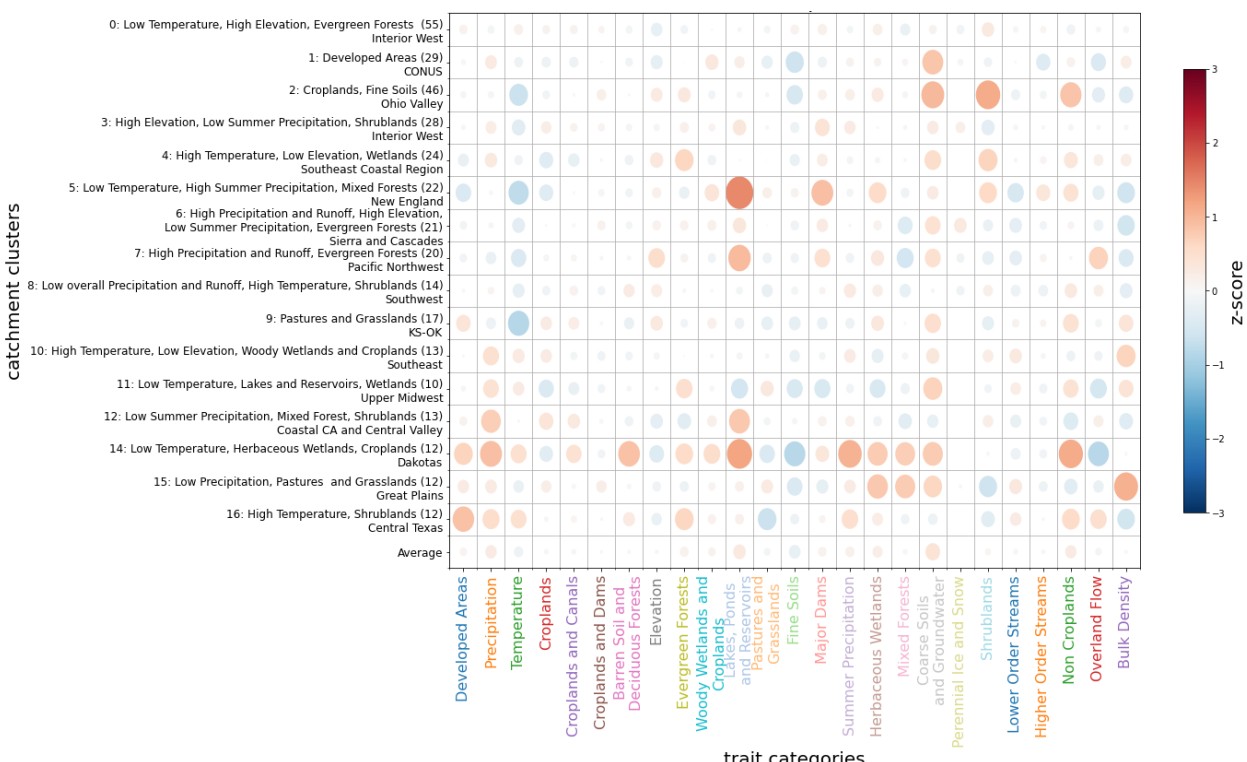

**Figure 15.** Bubble plot showing the z-scores of catchments where the baseflow is above the 90th percentile relative to the entire catchment cluster. The baseflow index is computed as the seven-day minimum flow divided by mean annual daily flows (averaged across all years). Bubble size is proportional to the absolute value of the z-score. Colors separate positive from negative values as indicated by the colorbar. Catchment clusters are displayed on the vertical axis using an identifier consistent with the one used in the original paper, a name describing their main characteristics, their approximate geographical area (if applicable), and the number of anomalous catchments above the 90th percentile shown in parenthesis in parenthesis. Only clusters with an anomalous set of catchments larger than 10 are included. Traits categories are displayed on the horizontal axis and are sorted in descending order, according to their size in terms of number of nodes in the traits network, and colored consistently with the trait clusters in said network. The last row of each plot refers to the average value of the trait z-scores of the clusters displayed in the plot and provides an idea of how much a trait category is over or under expressed across different clusters with different characteristics.

Overall, averaged z-scores for all catchments in the CONUS (shown in the last row of Fig. 15) indicates there is a moderate positive association of anomalously high base flows with the presence of lakes, ponds and reservoirs (11th column in light blue, Z=0.29), and with coarse soils and groundwater trait categories (18th column in gray, Z=0.40). Conversely, there is a negative link to fine soils (13th column in green, Z=-0.25). This indicates the potential for surface-groundwater exchange in regions where water bodies are present, and not surprisingly the importance of soil texture in mediating baseflow through infiltration and recharge.

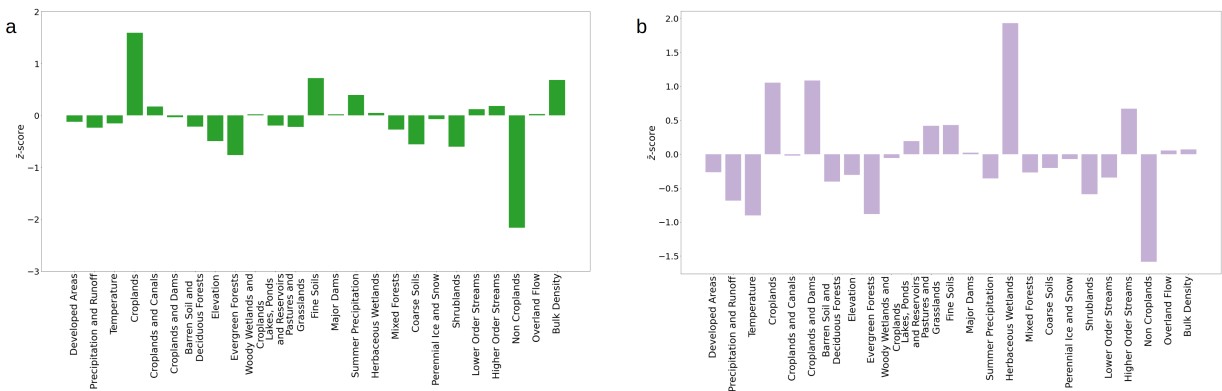

**Figure 16.** (a) Bar chart of traits z-scores of for the catchment cluster 2, characterized by croplands and fine soils. The catchments in this cluster are generally located in the Ohio Valley region. (b) Bar chart of traits z-scores of for the catchment cluster 14 , characterized by low temperatures, croplands and wetlands. The catchments in this cluster are generally located in North and South Dakota.

## 4.7    Application for regionalization studies

Our methodology can be used as initial steps of typical workflows used for predictions for unmonitored basins, which is (1) to classify catchments into groups for regionalization, and (2) to select a subset of traits from a large predictor space, a common challenge in many large sample studies. For the former, we choose to classify catchments using traits, since geospatial datasets are now available with a lot of trait information that allows us to do catchment classification at large spatial scales including for unmonitored catchments. For the latter, we find that we can condense a very large dataset containing hundreds of traits into

25 trait categories with the network approach, due to the redundancy in the traits. Thus our paired catchment-trait networks approach not only classifies catchments into clusters, but enables the reduction of a large dataset of traits into an interpretable set of trait categories by eliminating their redundancy. This provides the ability to identify distinct trait categories that are over- or under-expressed in catchment clusters to streamflow behaviors. The parallel analysis of cluster and traits data as networks is an important characteristic that distinguishes our method from other typical unsupervised clustering workflows.

## 5    Conclusions

In this study, we demonstrate a new network-based method for unsupervised classification of river catchments using their environmental and anthropogenic traits. This approach builds two parallel networks - the first identifies similar catchments, and the second reduces the large set of redundant traits into a small, interpretable set of trait categories. The method outperforms traditional unsupervised approaches for classification and enables simultaneous analysis of the catchment clusters and co-

expression of the trait categories. This makes it possible to identify the predominant traits that result in the partitioning of

catchments into different groups, as well as their distinct hydrologic behaviors. This trait-based approach can be used beyond hydrological applications to classify high dimensional datasets that have correlated information.

Using this method, we classified 9,067 catchments in the continental United States based on 274 traits from the GAGES-II dataset. The resulting clusters tend to be geographically coherent, even though no coordinates were used as inputs, revealing spatial patterns for catchments with predominantly natural or anthropogenic traits. The method can be implemented at multiple spatial scales for which trait data are available. As expected, the catchment clusters and associated trait expressions vary according to the spatial scale used for the analysis. We also find that the catchment clusters display distinct streamflow characteristics, as quantified by hydrological indices, indicating the potential for using the trait-based classification to understand and predict hydrological behavior.

We can also leverage network-specific metrics, like the degree centrality using this workflow. We demonstrate their utility by identifying a small set of watersheds that are representative for a group of catchments with specific traits. These representative sites can be used to determine locations to conduct observations or modeling activities that are transferable to other catchments.

Catchments are complex systems and their hydrological behavior is determined by the combined effect of multiple, co-dependent traits. Thus, networks are an extremely useful tool for investigating catchment properties and behavior, since they enable a holistic approach to the study of the system. The collective analysis of traits creates the opportunity to average out the unique characteristics of individual catchments, and characterize emergent hydrological behaviors across spatial scales.

*Code and data availability.* The daily observations of streamflow and specific conductance used in this study are publicly available from USGS NWIS at https://waterdata.usgs.gov/nwis. The characteristics of catchments from GAGES-II dataset are publicly available and can be accessed at https://pubs.er.usgs.gov/publication/70046617. The data used are publicly available with a CCBy4 license on the U.S. Department of Energy ESS-DIVE data repository and are accessible with the doi:10.15485/1987555 (Ciulla and Varadharajan, 2023). The code used will be released open source with a modified BSD 3-clause license as a Github repository, if this paper is accepted. The files for the codes are attached for review purposes with the manuscript submission.

## Appendix A: Data preprocessing

The traits in GAGES-II with prefix "RAW_" describe the straight line distance (in km) of water stations from points of interest such as dams, canals, etc. If no such point of interest is present in the basin, the trait is assigned a missing value of -999. This choice of missing values makes the trait non-monotonic because they have small positive values for stations in close proximity to a point of interest, large values for stations far from it, and a large negative value when the point is not present in the catchment. We transform these traits by computing the inverse of the variable for values greater than 0 (Fig. A1). The arbitrary -999 values are mapped into zeros. When gaging stations geographically coincide with the point of interest, the original zero distances are replaced with the maximum of the newly transformed values. In this way, the newly transformed trait values are monotonic and are a continuous and differentiable function of the distance.

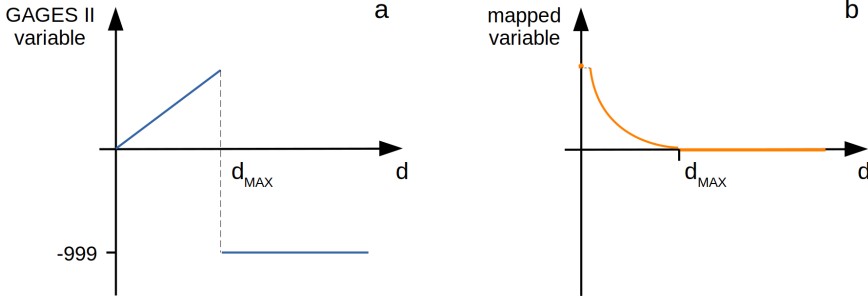

**Figure A1.** A pictorial representation of the trait mapping performed on a) the original variable denoting distance from a point of interest, and b) the resulting mapped values.

## Appendix B: Cao's method

The only free parameter of the PCA method is the number $k$, which is the final dimension of the reduced space. We determine $k$ using the false nearest neighbors method (FNN) from (Krakovská et al., 2015), which is based on the assumption that two points that are near each other in a sufficiently low dimensional space should remain close as the dimensions increase. In particular, we use the Cao's version of FNN (Cao, 1997), where the average distances $E(k+1)$ of the closest neighbor of all the elements in the space of dimension $k+1$ are divided by the same quantity computed in the $k$-dimensional space. For sufficiently high values of $k$, the ratio $R(k) = E(k+1)/E(k)$ approaches the value of 1, indicating that the average distances are not changing in two consecutive dimensions. We chose an arbitrary value of 1.05 to define the threshold below which the value $R(k)$ is assumed to be constant. The value $\hat{k}$ for which $R(\hat{k}) < 1.05$ determines the number of principal components to retain in the PCA method. For our dataset, the value of $\hat{k}$ is 20. By choosing $\hat{k} = 20$ dimensions in the reduced space, we account for 71.6% of the total variance of the original data.

## Appendix C: Cosine similarity

We present a comparison of the cosine similarity, our choice for the similarity measure, against the widely used euclidean distance and highlight why the latter is not the best choice to capture relationships within high-dimensional vectors.

The first reason is that vectors associated with two random points in an $n$-dimensional space have a high probability of being almost orthogonal when the number of dimensions $n$ diverges (Aggarwal et al., 2001). Thus:

$$\boldsymbol{x} \cdot \boldsymbol{y} \xrightarrow[n \to \infty]{} 0 \quad \boldsymbol{x}, \boldsymbol{y} \in \mathbb{R}^n \tag{C1}$$

The euclidean distance $D_E$ between two vectors $\boldsymbol{x}$ and $\boldsymbol{y}$ is:

$$D_E(\boldsymbol{x}, \boldsymbol{y}) = \sqrt{\sum_i^n (x_i - y_i)^2} = \sqrt{\|\boldsymbol{x}\|^2 + \|\boldsymbol{y}\|^2 - 2\boldsymbol{x} \cdot \boldsymbol{y}} \tag{C2}$$

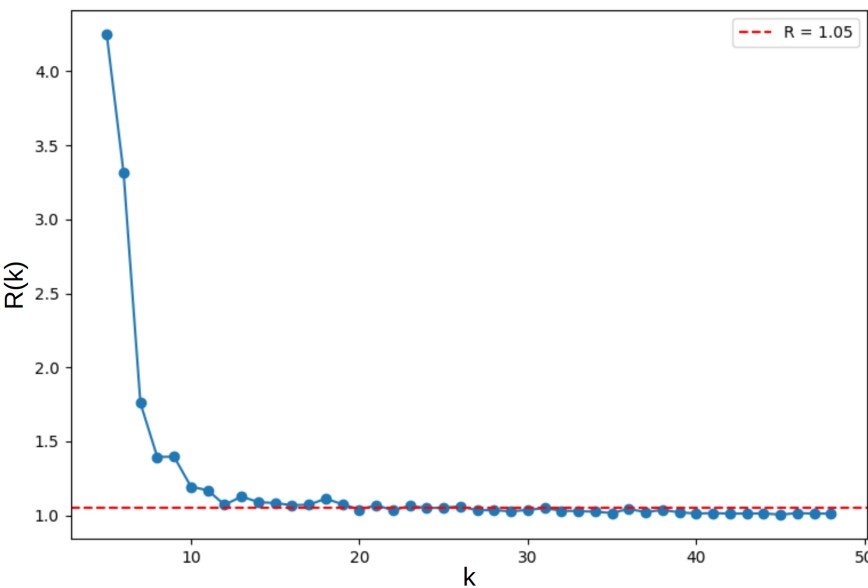

**Figure B1.** Ratio $R(k)$ as a function of the number of dimensions of the PCA. The horizontal dashed line in red indicates the arbitrary value $R = 1.05$.

where $\| \cdot \|$ represents the euclidean norm.

In the case of high-dimensional vectors, applying Equation C1 on Equation C2 causes the Euclidean distance to become a function of solely the vector norms:

$$D_E(\boldsymbol{x}, \boldsymbol{y}) \xrightarrow[n \to \infty]{} \sqrt{\|\boldsymbol{x}\|^2 + \|\boldsymbol{y}\|^2} \tag{C3}$$

This is not desirable because we lose the information about the relationship between the vectors, which is of interest when computing a distance metric. Although there is no definitive size threshold above which the effect of the high dimensionality will be prevalent, we expect that the effect would not be negligible even in the 20-dimensional space obtained after our dimensionality reduction step through the PCA.

Another consideration when choosing a distance measure is the type of information it provides. In the case of the Euclidean
distance, the directionality of the vectors associated with the data point does not influence the metric. Figure C1 illustrates an example in 2 dimensions where the directionality of the data is not captured by the euclidean distance. In particular, the points A and B have swapped coordinates ($x_A = y_B$ and $y_A = x_B$) with $x_A = y_B << y_A = x_B$. Instead C has the same B $y$ coordinate ($y_B = y_C$) and $x$ so that the euclidean distance between A and B $d_E(A, B)$, and B and C $d_E(B, C)$ are the same. So the Euclidean distance would indicate as equal the relationship between two points with swapped coordinates, which should
depict two very different elements, and between one of the two and a third one that shares exactly one coordinate, while the second coordinate is simply greater.

Conversely, the cosine distance captures the difference between A-B and B-C, because it considers the directionality. Specifically, the angle $\alpha_{AB}$ between A and B is much greater than the one $\alpha_{BC}$ between B and C leading to a bigger cosine distance. The cosine similarity captures the information contained in the different coordinates of the data points. In our methodology, the coordinates are the components given by the principal components as computed by the PCA, which represent a mixture of traits. We contend that in a trait based approach the directionality is important both before and after the dimensionality reduction and needs to be captured.

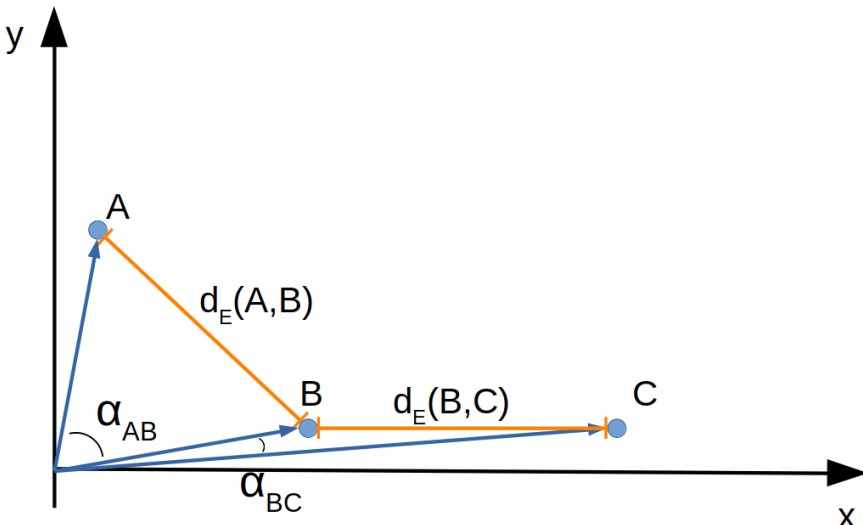

**Figure C1.** A pictorial representation of the qualitative difference between euclidean distance and cosine similarity. The point B is equidistant to A and C as per the euclidean distance metric even though its coordinate values are swapped with respect to A.

Hence, to mitigate the effects of dimensionality and include information on directionality, we use the cosine similarity instead of the Euclidean distance to compute the relationship between vectors. Computationally, the cosine similarity is computed as the dot product of a matrix with normalized row vectors and its transpose. In Equation 1, the dot product between the two vectors is the numerator of a ratio with the vector norms. Since it is a multiplicative factor, the dot product is retained regardless of how small its value is. For the Euclidean metric, a negligible value of the dot product will be outweighed by the norms of the vectors.

Finally, it is worth noting that the Euclidean distance and the cosine similarity are related to each other, and are equivalent when applied to normalized vectors. Specifically, when the vectors are normalized, then Equation C2 becomes

$$D_E = \sqrt{2(1 - S_C)} = \sqrt{2D_C} \tag{C4}$$

where $D_C = 1 - S_C$ is defined as the cosine distance.

**Appendix D: Brief Introduction to Networks**

A network $G$ is a mathematical object constituted by two sets $\{N, E\}$ where $N$ is the set containing the nodes of the network
and $E$ is the set of pairs of nodes called edges: $E = \{e = (a,b) s.t. a, b \in N\}$. A network is a great tool to model relationships
and interactions among the elements of a system (Börner et al., 2007). The elements of the system are represented as nodes
while the relationships are described by the edges. If the relationship between a pair of nodes is commutative, then the network
is called undirected, meaning that there is no directionality in the edge: $(a,b) = (b,a)$. Edges with the same node as endpoints
are self-loops, and may or may not be present according to what the edges represent. An edge can be simply present or not (for
which unweighted networks are appropriate), or it can carry a weight representing the strength of the edge whose interpretation
depends on the system under analysis. The maximum number of edges present in an undirected network with no self-loops, is
the number of possible combinations of pairs of elements in $N$ without considering the order $\binom{|N|}{2} = |N|(|N|-1)/2$. Here, $|N|$
is the number of nodes and also represents the size of the network. A network that has all possible edges is called complete,
and its edge number is $O(|N|^2)$. This rarely occurs for real world phenomena, and hence typically the number of edges in
real-world networks is much smaller than $O(|N|^2)$ (Newman, 2005). This is a crucial point, because it is the arrangement of
edges and their eventual heterogeneous distribution, namely the network topology, that gives rise to connectivity patterns that
reveal insights about the system. We refer to networks as being complex when they display non-trivial topological features,
with patterns of connection between their elements that are neither regular and yet not purely random. All this information is
condensed in an adjacency matrix, where both rows and columns are nodes, and each element of the matrix is the weight of
the edge between the two nodes identifying the matrix element and zero otherwise. If the network is undirected such a matrix
is symmetric. For a complete review of network properties refer to (Newman, 2018).

**Appendix E: Clustering Coefficient**

The clustering coefficient is a network measure that quantifies the connections among the neighbors of a certain node (Wasserman and Faust, 1994; Scott and Carrington, 2023). This measure ranges from zero (corresponding to the situation where no
neighbors of a node are connected to each other) to one (where all the neighbors of a node are connected to each other). We
compute the clustering coefficient for all the nodes in both the catchments and traits networks and show the results as a probability density function in E1. In both cases, the clustering coefficient spans the entire possible range of 0 to 1. In the network
of catchments the average clustering coefficient, computed by averaging the clustering coefficient of all nodes, is 0.61, while
in the network of traits it is equal to 0.75. This means that in both cases, on average, more than half of the possible connections
among neighbors are present. This fact indicates the presence of a group of nodes well connected with each other, and is the
first indication of the presence of complex connectivity patterns that can lead to the formation of clusters in both networks.

In the traits network the value with the highest probability is one, meaning that there are parts of the network that form
complete cliques, namely subsets of nodes that have all the possible edges allowed among them. This fact confirms the intuition
that some of the attributes provide redundant information, because their similarity, in terms of co-expression as learned by the
PCA, generates such tightly connected areas.

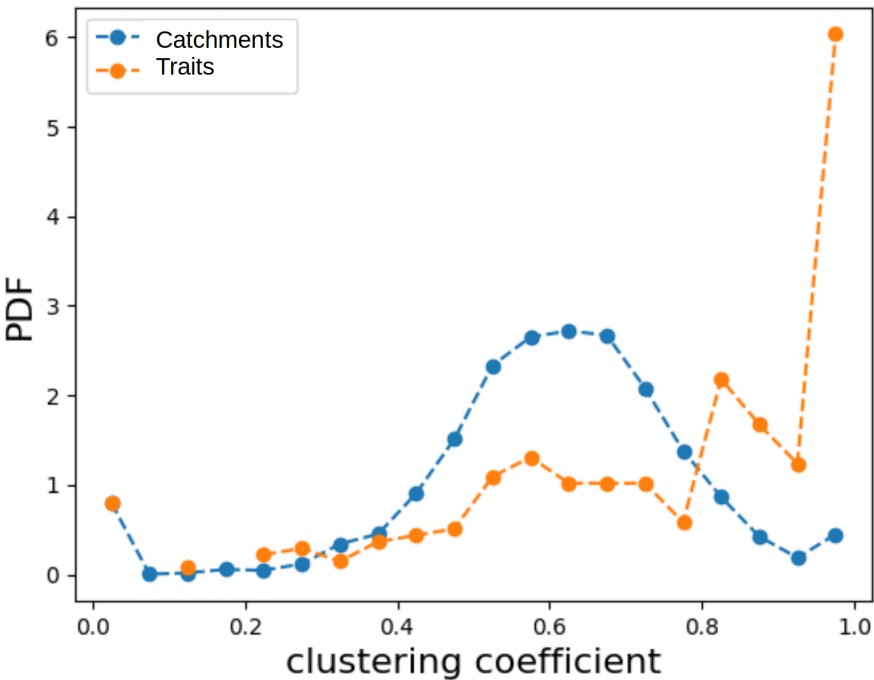

**Figure E1.** Probability distribution function of clustering coefficients for the network of catchments, in blue, and the network of traits, in orange.

## Appendix F: Traits Network Clusters Composition

| Cluster ID: 1 ; Cluster Name: Developed Areas ; Cluster Size: 35 ; Anthropogenic: yes |
|---|
| PDEN_2000_BLOCK, PDEN_NIGHT_LANDSCAN_2007, IMPNLCD06, PDEN_DAY_LANDSCAN_2007, DEVNLCD06, MAINS800_DEV, RIP100_DEV, RIP100_22, RIP800_DEV, DEVLOWNLCD06, MAINS800_22, RIP800_22, ROADS_KM_SQ_KM, MAINS100_22, MAINS100_DEV, RD_STR_INTERS, MAINS100_21, MAINS800_21, RIP100_21, DEVOPENNLCD06, RIP800_21, DEVMEDNLCD06, FRESHW_WITHDRAWAL, NLCD01_06_DEV, DEVHINLCD06, MAINS100_23, MAINS100_24, MAINS800_23, MAINS800_24, RIP100_23, RIP100_24, RIP800_23, RIP800_24, HGAC, DDENS_2009 |
| Cluster ID: 2 ; Cluster Name: Precipitation and Runoff ; Cluster Size: 30 ; Anthropogenic: no |

RUNAVE7100, WB5100_ANN_MM, NOV_PPT7100_CM, PPTAVG_BASIN, OCT_PPT7100_CM, WB5100_NOV_MM, DEC_PPT7100_CM, WD_SITE, WDMAX_BASIN, WB5100_MAY_MM, PPTAVG_SITE, WB5100_APR_MM, WB5100_JUN_MM, WD_BASIN, WDMAX_SITE, APR_PPT7100_CM, WB5100_OCT_MM, WB5100_DEC_MM, JAN_PPT7100_CM, FEB_PPT7100_CM, MAR_PPT7100_CM, WB5100_JAN_MM, WB5100_FEB_MM, WB5100_MAR_MM, WB5100_SEP_MM, WB5100_JUL_MM, WB5100_AUG_MM, RH_BASIN, RH_SITE, PRECIP_SEAS_IND

| Cluster ID: 3 ; Cluster Name: Temperature ; Cluster Size: 22 ; Anthropogenic: no |
| --- |

T_AVG_BASIN, T_AVG_SITE, T_MAX_BASIN, T_MAX_SITE, T_MIN_BASIN, T_MIN_SITE, PET, FEB_TMP7100_DEGC, MAR_TMP7100_DEGC, APR_TMP7100_DEGC, MAY_TMP7100_DEGC, SEP_TMP7100_DEGC, OCT_TMP7100_DEGC, NOV_TMP7100_DEGC, DEC_TMP7100_DEGC, FST32F_BASIN, AUG_TMP7100_DEGC, RFACT, JAN_TMP7100_DEGC, JUN_TMP7100_DEGC, JUL_TMP7100_DEGC, FST32SITE

| Cluster ID: 4 ; Cluster Name: Croplands ; Cluster Size: 22 ; Anthropogenic: yes |
| --- |

MAINS800_PLANT, PLANTNLCD06, MAINS100_PLANT, RIP100_PLANT, RIP800_PLANT, NITR_APP_KG_SQKM, PHOS_APP_KG_SQKM, MAINS100_82, CROPSNLCD06, MAINS800_82, RIP100_82, RIP800_82, CDL_CORN, CDL_SOYBEANS, FRAGUN_BASIN, PASTURENLCD06, MAINS100_81, MAINS800_81, RIP100_81, RIP800_81, CDL_OTHER_HAYS, HIRES_LENTIC_DENS

| Cluster ID: 5 ; Cluster Name: Croplands and Canals ; Cluster Size: 16 ; Anthropogenic: yes |
| --- |

RAW_DIS_NEAREST_CANAL, CANALS_PCT, CANALS_MAINSTEM_PCT, PCT_IRRIG_AG, CDL_RICE, PCT_NO_ORDER, NPDES_MAJ_DENS, PESTAPP_KG_SQKM, CDL_ORANGES, HYDRO_DISTURB_INDX, STREAMS_KM_SQ_KM, HGBD, RAW_DIS_NEAREST_MAJ_NPDES, ASPECT_NORTHNESS, RAW_AVG_DIS_ALLCANALS, RAW_AVG_DIS_ALL_MAJ_NPDES

| Cluster ID: 6 ; Cluster Name: Croplands and Dams ; Cluster Size: 15 ; Anthropogenic: yes |
| --- |

CDL_OTHER_CROPS, RAW_DIS_NEAREST_DAM, RAW_AVG_DIS_ALLDAMS, CDL_BARLEY, CDL_DURUM_WHEAT, CDL_DRY_BEANS, CDL_POTATOES, RAW_DIS_NEAREST_MAJ_DAM, RAW_AVG_DIS_ALL_MAJ_DAMS, MAJ_DDENS_2009, CDL_SPRING_WHEAT, CDL_OATS, HGBC, CDL_SUNFLOWERS, CDL_ALFALFA

| Cluster ID: 7 ; Cluster Name: Barren Soil and Deciduous Forests ; Cluster Size: 12 ; Anthropogenic: no |
| --- |

MINING92_PCT, PADCAT1_PCT_BASIN, BARRENNLCD06, DECIDNLCD06, MAINS100_31, MAINS800_31, RIP100_31, RIP800_31, RIP800_41, MAINS100_41, MAINS800_41, RIP100_41

| Cluster ID: 8 ; Cluster Name: Elevation ; Cluster Size: 12 ; Anthropogenic: no |
| --- |

| |
|---|
| ELEV_MAX_M_BASIN, ELEV_MEDIAN_M_BASIN, SNOW_PCT_PRECIP, ELEV_MEAN_M_BASIN, T_MINSTD_BASIN, LST32F_BASIN, ELEV_MIN_M_BASIN, ELEV_SITE_M, T_MAXSTD_BASIN, ELEV_STD_M_BASIN, LST32SITE, SLOPE_PCT |
| Cluster ID: 9 ; Cluster Name: Evergreen Forests ; Cluster Size: 12 ; Anthropogenic: no |
| PADCAT3_PCT_BASIN, MAINS100_42, MAINS100_FOREST, MAINS800_42, RIP100_FOREST, RIP100_42, MAINS800_FOREST, FORESTNLCD06, EVERGRNLCD06, RIP800_42, PADCAT2_PCT_BASIN, RIP800_FOREST |
| Cluster ID: 10 ; Cluster Name: Woody Wetlands and Croplands ; Cluster Size: 11 ; Anthropogenic: yes |
| MAINS100_90, TOPWET, RIP800_90, CDL_PEANUTS, ROCKDEPAVE, MAINS800_90, RIP100_90, WOODYWETNLCD06, PERDUN, CDL_WWHT_SOY_DBL_CROP, CDL_COTTON |
| Cluster ID: 11 ; Cluster Name: Lakes, Ponds and Reservoirs ; Cluster Size: 10 ; Anthropogenic: yes |
| ARTIFPATH_PCT, MAINS800_11, RIP100_11, STOR_NID_2009, STOR_NOR_2009, HIRES_LENTIC_PCT, HIRES_LENTIC_MEANSIZ, WATERNLCD06, RIP800_11, MAINS100_11 |
| Cluster ID: 12 ; Cluster Name: Pastures and Grasslands ; Cluster Size: 10 ; Anthropogenic: yes |
| CDL_FALLOW_IDLE, GRASSNLCD06, MAINS100_71, MAINS800_71, RIP100_71, RIP800_71, CDL_SORGHUM, CDL_WINTER_WHEAT, CDL_PASTURE_GRASS, ASPECT_EASTNESS |
| Cluster ID: 13 ; Cluster Name: Fine Soils ; Cluster Size: 10 ; Anthropogenic: no |
| AWCAVE, CONTACT, NO200AVE, SILTAVE, NO10AVE, KFACT_UP, NO4AVE, CLAYAVE, HGC, HGD |
| Cluster ID: 14 ; Cluster Name: Major Dams ; Cluster Size: 9 ; Anthropogenic: yes |
| ARTIFPATH_MAINSTEM_PCT, PCT_6TH_ORDER_OR_MORE, DRAIN_SQKM, NDAMS_2009, POWER_NUM_PTS, MAJ_NDAMS_2009, POWER_SUM_MW, HIRES_LENTIC_NUM, STRAHLER_MAX |
| Cluster ID: 15 ; Cluster Name: Summer Precipitation ; Cluster Size: 7 ; Anthropogenic: no |
| SEP_PPT7100_CM, WDMIN_BASIN, WDMIN_SITE, MAY_PPT7100_CM, JUN_PPT7100_CM, JUL_PPT7100_CM, AUG_PPT7100_CM |
| Cluster ID: 16 ; Cluster Name: Herbaceous Wetlands ; Cluster Size: 7 ; Anthropogenic: no |
| HGAD, OMAVE, MAINS100_95, RIP100_95, RIP800_95, EMERGWETNLCD06, MAINS800_95 |
| Cluster ID: 17 ; Cluster Name: Mixed Forests ; Cluster Size: 6 ; Anthropogenic: no |
| HGCD, MIXEDFORNLCD06, MAINS100_43, MAINS800_43, RIP100_43, RIP800_43 |
| Cluster ID: 18 ; Cluster Name: Coarse Soils ; Cluster Size: 6 ; Anthropogenic: no |
| BFI_AVE, HGA, PERMAVE, HGB, SANDAVE, WTDEPAVE |
| Cluster ID: 19 ; Cluster Name: Perennial Ice and Snow ; Cluster Size: 5 ; Anthropogenic: no |

| SNOWICENLCD06, MAINS100_12, MAINS800_12, RIP100_12, RIP800_12 |
| --- |
| Cluster ID: 20 ; Cluster Name: Shrublands ; Cluster Size: 5 ; Anthropogenic: yes |
| SHRUBNLCD06, MAINS100_52, MAINS800_52, RIP100_52, RIP800_52 |
| Cluster ID: 21 ; Cluster Name: Lower Order Streams ; Cluster Size: 5 ; Anthropogenic: no |
| PCT_1ST_ORDER, RRMEDIAN, BAS_COMPACTNESS, PCT_2ND_ORDER, RRMEAN |
| Cluster ID: 22 ; Cluster Name: Higher Order Streams ; Cluster Size: 4 ; Anthropogenic: no |
| PCT_3RD_ORDER, MAINSTEM_SINUOUSITY, PCT_5TH_ORDER, PCT_4TH_ORDER |
| Cluster ID: 23 ; Cluster Name: Non Croplands ; Cluster Size: 1 ; Anthropogenic: yes |
| CDL_ALL_OTHER_LAND |
| Cluster ID: 24 ; Cluster Name: Overland Flow ; Cluster Size: 1 ; Anthropogenic: no |
| PERHOR |
| Cluster ID: 25 ; Cluster Name: Bulk Density ; Cluster Size: 1 ; Anthropogenic: no |
| BDAVE |

Table F1: List of traits aggregated by the traits network clusters. For each cluster we provide an header including a unique ID, a representative name, the size of the cluster in terms of nodes, indicate if the cluster includes anthropogenic factors, and the list of traits in the cluster.

## Appendix G: Hydrological Indices Statistics

| Hydro-logical index | Description | K-W test | K–S test one-sample | K–S test two-samples |
| --- | --- | --- | --- | --- |
| ma41 | Mean annual flow divided by catchment area (mm/day) | True | 97.06 | 95.19 |
| dh13 | Mean annual of 30-day maximum divided by median flow | True | 94.12 | 90.55 |
| mh14 | Median of the highest annual daily flow divided by the median annual daily flow | True | 97.06 | 90.37 |
| ma5 | Skewness in daily flow | True | 91.18 | 90.20 |
| mh16 | Mean of the 10th percentile from the flow duration curve divided by median daily flow across all years | True | 91.18 | 88.95 |
| ma3 | Coefficient of variation in daily flows | True | 94.12 | 88.95 |
| fh7 | Mean number of high flow events per year using an upper threshold of 7 times the median flow over all years | True | 94.12 | 87.88 |

| Hydro-logical index | Description | K-W test | K–S test one-sample | K–S test two-samples |
|---|---|---|---|---|
| ra6 | Median of difference between natural logarithm of flows between two consecutive days with increasing flow | True | 88.24 | 86.27 |
| fh6 | Mean number of high flow events per year using an upper threshold of 3 times the median flow over all years | True | 100.00 | 85.92 |
| th3 | Maximum proportion of the year (number of days/365) during which no floods have ever occurred over the period of record | True | 85.29 | 85.20 |
| fl3 | Total number of low flow spells (threshold equal to 5% of mean daily flow) | True | 82.35 | 83.60 |
| fh3 | High flood pulse count (high flood: at least 3 times median of daily flows) | True | 85.29 | 83.07 |
| ml21 | Coefficient of variation in annual minimum flows averaged across all years | True | 85.29 | 81.46 |
| ml17 | Seven-day minimum flow divided by mean annual daily flows averaged across all years | True | 94.12 | 81.28 |
| dh16 | Coefficient of variation of high flood pulse (high flood: at least 75th percentile of daily flows) | True | 91.18 | 80.93 |
| dl13 | Mean annual of 30-day minimum divided by median flow | True | 79.41 | 80.39 |
| fh2 | Coefficient of variation of high flood pulse count (high flood: at least 75th percentile of daily flows) | True | 88.24 | 78.97 |
| ml18 | Coefficient of variation of seven-day minimum flow divided by mean annual daily flows averaged across all years | True | 76.47 | 78.25 |
| fl2 | Coefficient of variation of low flood pulse count | True | 94.12 | 77.36 |
| ta1 | Constancy | True | 79.41 | 77.36 |
| ml4 | Mean minimum monthly flow for the months of April (m3/s) | True | 79.41 | 77.36 |
| dh20 | Mean duration of high flood pulse (high flood: at least 25th percentile of median flows) (days) | True | 85.29 | 77.18 |
| mh10 | Mean maximum flows for the months of October (m3/s) | True | 94.12 | 76.47 |
| tl2 | Variability in Julian date of annual minimum (days) | True | 82.35 | 76.11 |
| mh8 | Mean maximum flows for the months of August (m3/s) | True | 85.29 | 75.94 |
| ma11 | Spread in 75th-25th percentile range on decimal logarithm transformed daily flows | True | 76.47 | 75.58 |
| dl17 | Coefficient of variation of low flood pulse count | True | 79.41 | 75.22 |

| Hydro-logical index | Description | K-W test | K–S test one-sample | K–S test two-samples |
|---|---|---|---|---|
| dh15 | Mean duration of high flood pulse (high flood: at least 75th percentile of daily flows) (days) | True | 79.41 | 74.15 |
| ra8 | Number of negative and positive changes in water conditions from one day to the next | True | 70.59 | 73.98 |
| ra5 | Ratio of days where the flow is higher than the previous day | True | 67.65 | 67.02 |
| ra9 | Coefficient of variation of the number of negative and positive changes in water conditions from one day to the next | True | 64.71 | 62.75 |
| fl1 | Low flood pulse count (low flood: below 25th percentile of daily flows) | True | 61.76 | 59.36 |
| dl18 | Number of zero-flow days (days) | True | 70.59 | 56.33 |
| dl16 | Mean duration of low flood pulse (days) | True | 47.06 | 50.98 |

Table G1: Table showing the results from statistical tests comparing the distributions of streamflow indices of catchment clusters resulting from our network-based methodology. The first two columns list the streamflow indices used in this study as alphanumeric codes with brief descriptions as in Olden and Poff (2003). The last 3 columns show the result of the Kruskall-Wallis (K-W) test indicating that not all the samples have the same distribution, 1-sample and 2-sample Kolmogorov-Smirnov (K-S) tests. The indices are sorted according to the 2-sample Kolmogorov-Smirnov Test in descending order.

# Appendix H: Comparison with Traditional Unsupervised Techniques

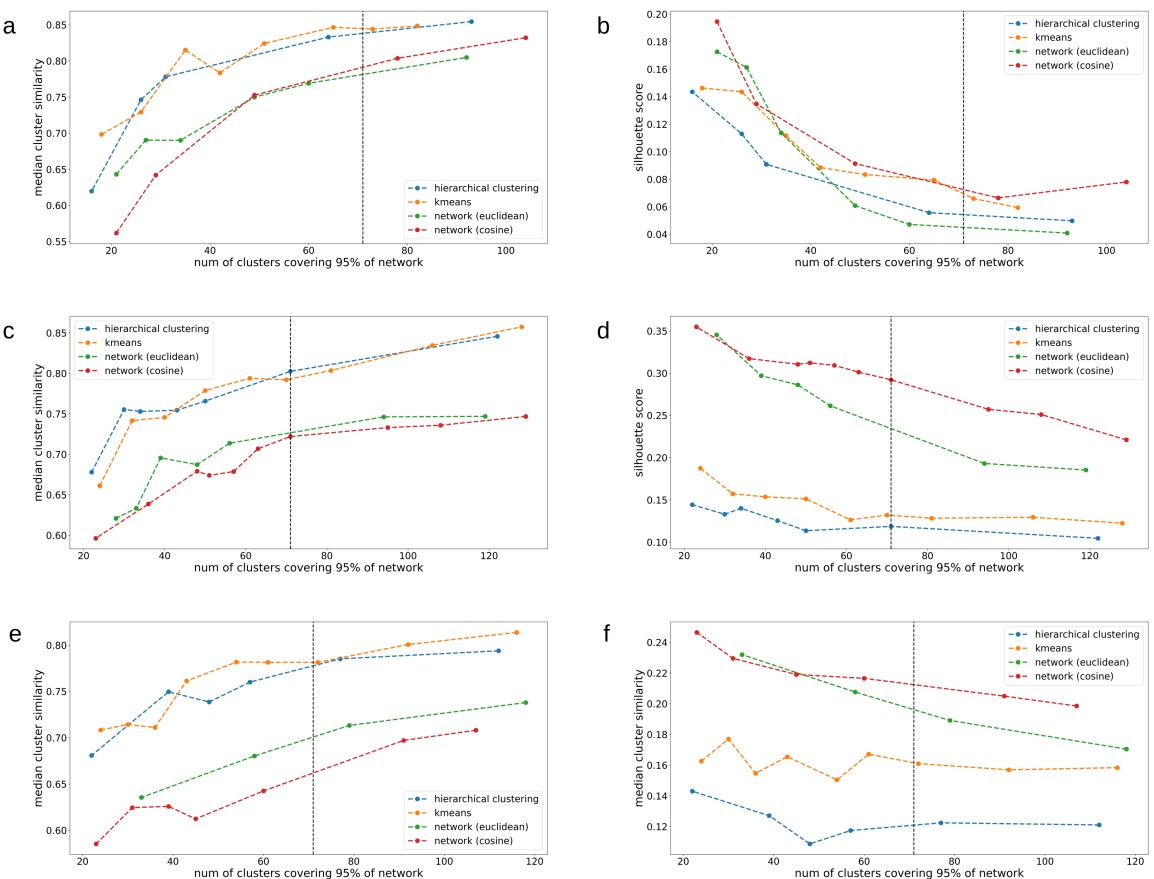

**Figure H1.** Median cluster similarity values (a, c, e) and silhouette scores (b, d, f) for different clustering methods and similarity measures used in the network analysis. The number of reduced dimensions after PCA is equal to (a, b) 6, (c, d) 20 (used in the study) and (e,f) 90 corresponding to 50%, 72% and 95% of retained information respectively. The vertical black dashed line refers to the cluster granularity used in the paper. The colored dashed lines are shown for visualization of trends. Lower values of the median cluster similarity metric (a, c, e) correspond to better clustering performance. Higher values of the silhouette scores (b, d, f) correspond to better clustering performance.

## Appendix I: Hydrological Indices Correlation with Traits

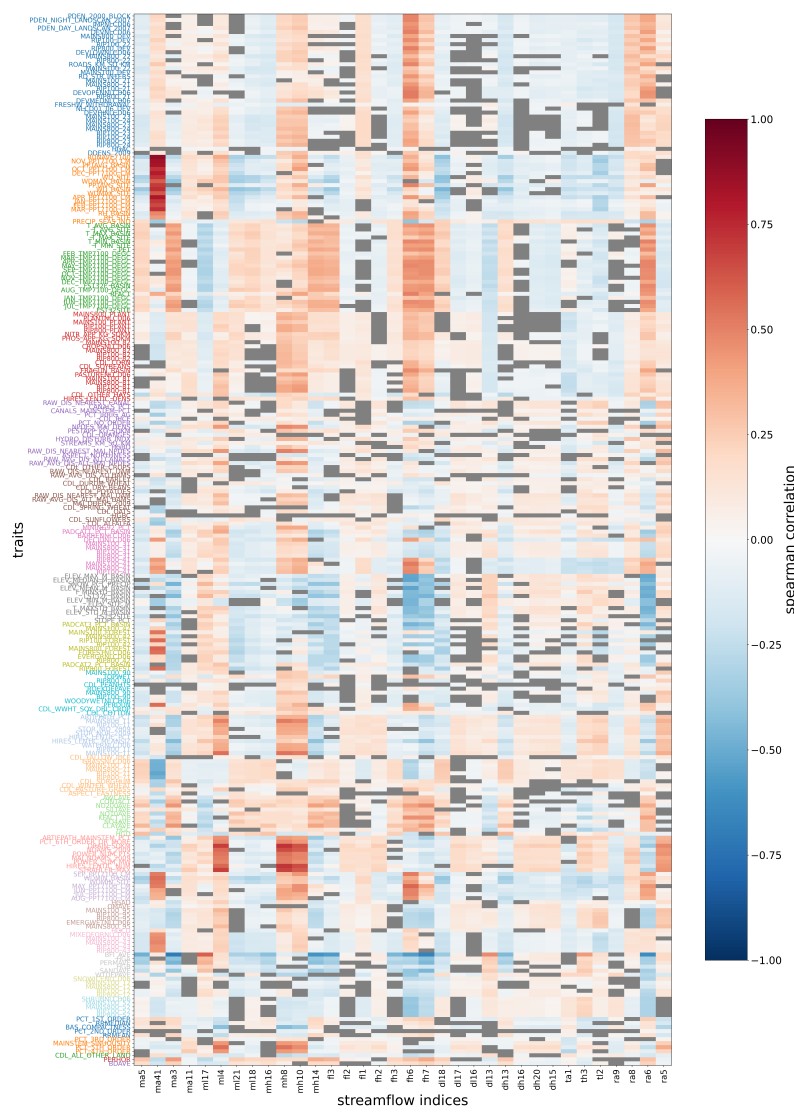

**Figure I1.** Heatmap showing spearman correlation coefficients between streamflow indices and the traits used in the study. The intensity of the colors show the degree of correlation or anticorrelation as indicated in the color bar. Traits on the horizontal axis are ordered and colored according to the traits categories they belong to. Gray boxes indicate correlations that are above the significance level of p>0.05. The primary purpose of this plot is to provide a visual representation of the redundancy in the correlations between the traits within the same category and streamflow indices.

*Author contributions.* Author contributions follow the CRediT taxonomy. Conceptualization: F.C., C.V., Data curation: C.V., Formal Analysis: F.C., Funding Acquisition: C.V., Investigation: F.C., Methodology: F.C., Project Administration: C.V., Resources: C.V., Software: F.C., Supervision: C.V., Validation: F.C., C.V., Visualization: F.C., Writing - original draft: F.C., Writing - review and editing: C.V.

*Competing interests.* The authors declare that they have no known competing financial interests or personal relationships that could have appeared to influence the work reported in this paper.

*Acknowledgements.* This research is supported by the Early Career Research Program funded by the U.S. Department of Energy, Office of Science, Office of Biological and Environmental Research under the Berkeley Lab Contract Number DE-AC02-05CH11231. We acknowledge the discussions with Eoin Brodie on trait identification, and group discussions with Helen Weierbach, Mohammed Ombadi, Aranildo Lima, and Jared Willard, which inspired ideas for inclusion in this paper and provided feedback on the methodology. We also acknowledge Valerie Hendrix and Danielle Christianson for their help with using the BASIN-3D software to download the data used in this paper.

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
