# Peer review of "A Network Approach for Multiscale Catchment Classification using Traits"

_EGUsphere, 2023_

## Author Comment (AC1)

**Manuscript title**
A Network Approach for Multiscale Catchment Classification using Traits
**Authors**
Fabio Ciulla and Charuleka Varadharajan

**RC1**

Major comment

COMMENT #1: The authors introduce a novel method to cluster catchments that is based on traits. The dataset is impressive and the network-based classification is, to my understanding, a relevant and innovative approach in this case. Methods and results are well presented.

AUTHOR RESPONSE #1 - We thank the reviewer for the positive comments. The reviewer brings up important concerns, and we have provided detailed responses below for each concern that was raised. If invited to resubmit, we would make changes that address these comments, and believe that it would significantly improve the paper.

COMMENT #2 - My main concern with such unsupervised classification is how we can use it for practical hydrological studies.

AUTHOR RESPONSE #2: We agree with the comments from both reviewers that we could do more to demonstrate the utility of our methodology for hydrological applications. In addition to responding to comments from reviewer 1 below, we note that we have done additional analysis to show how we can gain hydrological insights from our approach in our response to comments from reviewer 2.

COMMENT #3: From the introduction and discussion, it appears one aim of clustering is application to ungauged basins. In this sense, the results of the paper are discouraging, because the clustering technique does not succeed in relating 'traits' clusters to hydrological behaviors, except for some specific hydrological traits. This part is essential, in my opinion, for switching from a mere clustering exercise to something which could actually be useful in hydrological practice. I do not know how the method can be tuned to improve the overlap between the geographical and hydrological clusters, but my wish is that the authors tackle this issue in the paper. I realize that this implies a significant change in the paper.

AUTHOR RESPONSE #3: As pointed out,one of the applications of our methodology is for predictions in unmonitored basins (PUBs). We had originally demonstrated that our approach to trait-based clustering (which the reviewer refers to as geographical clustering) results in statistically distinct hydrological behavior (based on signatures) relative to other catchments within the CONUS. We assume the reviewer thinks the results are

discouraging, based on the boxplots shown in Figure 13 in the original manuscript, where it appears as though some of the distributions overlap for the two indices shown. However, we would like to point out that our current approach does result in distinguishing the streamflow behavior for most of the indices, indicating that there already is significant overlap between the trait-based clustering and the hydrological groupings. To demonstrate this overlap, we conducted two additional statistical tests that further indicate that the hydrological indices are mostly distinct between the catchment clusters.

The first is a nonparametric 1-sample Kolmogorov-Smirnov Test that compares a sample distribution to a reference one for each hydrological index. This expands on the Kruskall-Wallis test that we presented in the manuscript, but allows us to determine the number of clusters that are statistically distinct from the entire catchment dataset. Here each sample is constituted by the indices of one cluster and the reference distribution is based on all the catchments within the CONUS. The null hypothesis is that samples are drawn from the reference distribution when using 0.05 as threshold for the p-value. We find that on average 83% of the clusters reject the null hypothesis, meaning that the distribution of their streamflow indices is distinct most of the time from the overall distribution at the continental scale.

The second is a 2-sample Kolmogorov-Smirnov test comparing the distributions of indices for pairs of catchment clusters, which allows us to determine how different the clusters are from each other. Here each sample pair is constituted by the distribution of indices for the clusters being compared. Similar to the one-sample test, the null hypothesis is that samples are drawn from the same distribution when using 0.05 as threshold for the p-value. On average 79% of all the tests reject the hypothesis that the aggregated indices from the two clusters are drawn from the same distribution confirming that the trait-based classification generally results in distinct streamflow index distributions.

The results of all the statistical tests are summarized in Table RC-1 below for all 34 streamflow indices. These results show that the trait-based clustering approach results in distinct signature classification (i.e. the hydrological clustering), and that our methodology is a valid tool to partition catchments and investigate PUBs.

[revised manuscript text omitted]

Specifically regarding the statements *"the clustering technique does not succeed in relating 'traits' clusters to hydrological behaviors, except for some specific hydrological traits. This part is essential, in my opinion, for switching from a mere clustering exercise to something*

*which could actually be useful in hydrological practice."* , we note that since the dataset of traits is large, and encompasses a broad set of categories (climate, geology, land use, human activities etc.), it is expected that not all traits (or trait clusters) are going to be related to hydrologic behavior. The question about which traits are most relevant for a particular hydrologic function of interest, such as streamflows, is still largely unresolved. In general, prior large sample studies such as (Eng et al., 2017) and (Addor et al., 2018) have not had much success in using traits to predict hydrologic signatures with statistical or classical machine learning approaches. Tackling the issue of relating trait clusters to hydrologic behavior is a different study that is out of scope for this paper. This is the subject of multiple follow-on studies and papers that we are working on, which require building models to show the relationships between the trait clusters and hydrologic signatures. Please see our response #4 below where we elaborate more on the practical uses for our approach.

COMMENT #4: In the case the authors stick to unsupervised clustering, I guess that the paper might be of interest, but in my opinion, the authors should:

- introduce in more details the practical implications of such clustering, and

AUTHOR RESPONSE #4: In this paper, we focus on using the clustering for the first few steps of the typical workflows used in PUB studies, which is (1) to classify catchments into groups for regionalization, and (2) to select a subset of traits from a large predictor space, which is a common challenge in many large sample studies. For the former, we choose to classify catchments using traits, since geospatial datasets are now available with a lot of trait information that allows us to do catchment classification at large spatial scales including for unmonitored catchments. For the latter, we find that we can condense a very large dataset of >274 traits into 25 trait categories with the network approach, due to the redundancy in the traits.

Thus our paired catchment-trait networks approach not only classifies catchments into clusters, but enables the reduction of a large dataset of traits into an interpretable set of trait categories by eliminating their redundancy. This provides the ability to identify distinct trait categories that are over- or under-expressed in catchment clusters to streamflow behaviors. The parallel analysis of cluster and traits data as networks is an important characteristic that distinguishes our method from other typical unsupervised clustering workflows.

In Figure 1 below, we illustrate how our approach helps with reducing the trait redundancy. In Fig 1, the spearman correlation coefficients ($\rho$) between streamflow indices on the vertical axis and catchment traits on the horizontal axis are shown, which account for non linearities in the data. Traits are sorted and colored according to the cluster they belong to in the traits network. In general, we find that traits belonging to the same cluster - i.e. traits categories as per our definition - have similar correlations with the streamflow indices, as indicated by the contiguous areas with similar colors for each row. In Figure 2, we show the

spearman correlation coefficients between the streamflow indices and catchment traits aggregated as a median on the trait categories generated in our method. Comparing Figures 1 & 2, we see that the relationships between hydrological signatures and several of the redundant traits are effectively similar, and hence our trait categories are sufficient to determine the relationship between traits and signatures.

[Figure]

Figure 1. Heatmap showing spearman correlation coefficients between streamflow indices and the traits used in the study. The intensity of the colors show the degree of correlation or anticorrelation as indicated in the color bar. Traits on the horizontal axis are ordered and colored according to the traits categories they belong to. Gray boxes indicate correlations that are below the significance level of $p<0.05$. The

primary purpose of this plot is to provide a visual representation of the redundancy in the correlations between the traits within the same category and streamflow indices.

[Figure]

Figure 2. Heatmap showing spearman correlation coefficients between streamflow indices and the traits categories generated in the study. The intensity of the colors show the degree of correlation or anticorrelation as indicated in the color bar. Gray boxes indicate correlations that are below the significance level of p<0.05.

Elaborating further on the examples described in our paper in Section 4.5, we can already see from this reduced set of trait categories in Figure 2, that across all 9067 catchments, mean annual runoff (ma41) is not just positively correlated with traits related to precipitation, but also with the presence of mixed forests ($\rho$=0.45) and to a lesser degree evergreen

forests ($\rho$=0.25). The ma41 index is also negatively correlated with the pastures and grasslands trait category ($\rho$=-.40). This highlights the general role of vegetation in mediating flows, and is somewhat counterintuitive given that in the absence of management, forested catchments with higher evapotranspiration would be expected to have lower flows compared to grasslands. As shown in Fig 13 of our paper, the catchments where there tends to be higher mean annual runoff is cluster 7, which is in the Pacific Northwest basin.

The fh6 index indicating the mean number of moderate floods per year (>3 times median flows) is not just positively correlated with precipitation traits, but also with traits related to developed areas ($\rho$=0.40), croplands ($\rho$=0.32) and temperature ($\rho$=0.43). The FH6 index is inversely correlated with elevation ($\rho$=-0.45), presence of shrublands ($\rho$=-0.38), evergreen forests ($\rho$=-0.28), coarse soils & groundwater ($\rho$=-0.35). These relationships are consistent across other flood indices. For example, the FH7 index showing the propensity for heavy floods (7 times median flows) similarly has a moderate positive correlation with temperature ($\rho$=0.44) and overland flow ($\rho$=0.38), and a moderate negative correlation with elevation ($\rho$=-0.39) and coarse soils/groundwater ($\rho$=-0.43). This indicates how flooding is affected by the complex relationships between land use, vegetation, soil infiltration capacity and base flows.

Hence we argue that the trait categories generated using our method are interpretable in a manner that is harder to do with a dimensionality reduction approach using Eigen vectors, where the contributions of traits can be distributed across many principal components. For additional applications of our unsupervised clustering approach, we refer the reviewer to our response #3 to reviewer 2 comments. Here we conducted additional analysis to identify relationships between hydrologic signatures and the catchment clusters, as well as the predominant traits of the catchments in those clusters. This analysis has produced new insights that on its own can be used to generate hypotheses about processes that influence hydrological behavior.

COMMENT #5:

- compare the obtained classification with a benchmark clustering approach.

AUTHOR RESPONSE #5: We acknowledge that the manuscript we submitted does not provide sufficient evidence that our proposed method based on networks and cosine similarity performs better than traditional unsupervised clustering algorithms. We thank the reviewer for this suggestion and in response performed a more comprehensive analysis comparing the performance of our method against benchmark hierarchical clustering and k-means approaches.

We reiterate that because of the lack of benchmark datasets, it is not straightforward to evaluate the performance of unsupervised methodologies because of the lack of classified

target variables.  In the absence of a benchmark dataset, we identified two metrics to evaluate the performance of the different methods.

The first metric, which we refer to as the "cluster similarly metric" has been already introduced in the paper (lines 459-472) and reflects the similarity in traits of the catchment clusters. Clusters can be represented by vectors using the average trait z-scores aggregated among the catchments belonging to each cluster. In this way each catchment cluster can be compared with others by calculating the pairwise cosine similarity. For each catchment cluster, the highest value of the similarity is used as a conservative measure of inter cluster similarity, for the purpose of assessing how far apart the clusters are from each other. The median of all the highest similarities represent how distinct the clusters produced by each algorithm are. A good algorithm should produce distinct clusters, so we aim to *minimize this metric*.

The second metric that we now added to the analysis is the silhouette score (Rousseeuw, 1987), which  measures how similar each element (i.e., a catchment) is to the cluster it belongs to with respect to the other clusters. The values of this metric range between -1 and 1, with higher values denoting that an element is well placed in its cluster compared to other clusters. The silhouette values are averaged for all  the items in the dataset. A good clustering algorithm would produce *higher values of the silhouette score*.

We use these two metrics to compare our clustering approach based on networks and cosine similarity against the hierarchical clustering (in its common implementation using the ward criterion) and the k-means clustering algorithm. Additionally, we also compare our method against a version where the pairwise similarity between nodes is computed using the euclidean distance instead of the cosine distance. This is done to show the difference produced by the metric choice while keeping the rest of the workflow unmodified. Finally, to show the robustness of our approach, we extended the comparison by exploring the landscape of the two free parameters in our workflow, namely the number of reduced dimensions after the PCA (k) and the cluster granularity. This last quantity is governed by different parameters according to the clustering method used. Three different values of k are investigated; k=6 corresponding to 50% of retained information after PCA, k=20 (our choice in the study) corresponding to 72% of retained information, and k=90 corresponding to 95% of retained information. For each value of k we generated clusters with the different methods so that the number of clusters covering 95% of the dataset ranges between 20 and 120.

The results of this investigation are shown in Fig. 3, which displays the cluster similarity metric, and Fig. 4, showing the silhouette score. According to these two metrics the performance of our network based clustering (red and green points) is considerably superior to both k-means (yellow points) and hierarchical clustering (blue points) across the different values of k and cluster granularity. This is evident from the consistently low values of the median cluster similarity and higher values of silhouette scores. Also, the network generated using the cosine distance as a similarity metric (red points) performs better than its counterpart that uses the euclidean distance (green points). This confirms that the cosine

similarity should be preferred as a metric distance in our investigation, where the dimensionality of the problem can be high and the directionality of the data carries valuable information.

If invited to resubmit, we would include this additional analysis as part of the "Methods", "Results" and "Discussion" sections.

[Figure]

Figure 3. Median cluster similarity values for different clustering methods and similarity measures used in the network analysis. The number of reduced dimensions after PCA is equal to (a) 6, (b) 20 (used in the paper) and (c) 90 corresponding to 50%, 72% and 95% of retained information respectively. The vertical black dashed line in (b) refers to the cluster granularity used in the paper. The colored dashed lines are shown for visualization of trends. Lower values of the median cluster similarity metric correspond to better clustering performance.

[Figure]

Figure 4. Silhouette scores for different clustering methods and similarity measures used in the network analysis. The number of reduced dimensions after PCA is equal to (a) 6, (b) 20 (used in the paper) and (c) 90 corresponding to 50%, 72% and 95% of retained information respectively. The vertical black dashed line in (b) refers to the cluster granularity used in the paper. The colored dashed lines are shown for visualization of trends. Higher values of the silhouette scores correspond to better clustering performance.

Minor comments

COMMENT #6: l.5: please clarify the term "subject to degradation"

AUTHOR RESPONSE #6: The term "degradation" is often used in computational literature dealing with metrics in high dimensions and refers to the property of a distance metric to perform worse as the number of dimensions grows. This concept of "degradation" is related to the "curse of dimensionality". It can be understood by recognizing that, counter to our intuition, what applies in three dimensions does not necessarily hold in higher dimensions. For example, in high dimensions, most of the mass of the points distributed according to a well behaved Gaussian distribution does not lie around the mean but becomes increasingly distant from it. Most of the mass migrates toward the surface of the domain leaving the bulk of the inner space empty. One of the consequences of this increased sparsity in high dimensional space is that the ratio of the distances of the nearest and farthest neighbors to a given point is almost 1, namely the points become uniformly distant from each other (Beyer et al., 1999).

We apologize for the lack of clarity and can explain the term when it occurs in the manuscript.

COMMENT #7: l.43, l.48 and in many other places: problems with in-line referencing.

AUTHOR RESPONSE #7: Thanks for pointing that out. We will correct the references if invited to submit a revised manuscript.

COMMENT #8: Section 2.3: I understand that traits values are standardized, but are their distributions normal? I guess no and I wonder how this may affect PCA and low dimensional vectors extracted from PCA.

AUTHOR RESPONSE #8: The referee's intuition about the non-normality behavior of the traits distribution is correct. The Shapiro-Wilk test, which checks if the data is drawn from a normal distribution, reveals that none of the traits are normally distributed when testing with a p-value of 0.05. However, the PCA method does not require normality in the input data. Instead, it is the presence of outliers that is detrimental to the PCA because it can unfairly promote dimensionality reduction toward directions associated with variables with outliers. We verified that only 8.1% of the traits lies outside their "inner fence", a common threshold for outliers, defined as the range between $Q1 - 1.5 * IQR$ and $Q3 + 1.5 * IQR$ for each trait, where $Q1$ and $Q3$ are the first and third quartiles respectively, and $IQR = Q3 - Q1$ is the interquartile range. Although this amount is not negligible, and given that there is not an universal threshold to consider a dataset compromised by outliers, we believe that the use of PCA is a reasonable choice.

COMMENT #9: l.473-475: Please clarify the added values of the network-based approach compared to other clustering techniques. Many of them address already the problem of dimensionality by working on Eigen-vectors.

AUTHOR RESPONSE #9: In response #4, we explain some of the benefits of our clustering approach. As shown in response #5, we have demonstrated that the network-approach outperforms other clustering techniques. Specific to the comment regarding the use of Eigen-vectors, we point out that although working with eigenvectors reduces the dimensionality of the problem, often the reduced vector space is still high dimensional. In our case, retaining 72% of the information - using the variance explained by the SVD singular values matrix - from the dimensionality reduction, still leads to a 20 dimension vector space. Although there is no universally accepted threshold for "high-dimensional" data, we argue that 20 constitutes a high number of dimensions where distance calculations are impacted.

Using a network approach allows one to choose the similarity metric and not to rely on euclidean distance, a metric needed in most of the traditional unsupervised clustering methods like k-means and hierarchical clustering. Thus we are able to use the cosine similarity metric that is less affected by issues of high dimensionality and include directionality information in the data.

The resulting clusters from our network approach are computed using the information from both the transformed matrix and the principal components. As illustrated in response #4, we believe that our workflow produces more interpretable results than the ones that would be obtained using only the Eigenvectors obtained from a PCA. This is because the network approach allows to separate traits or catchments into distinct clusters, produced using network connections statistics validated by the disparity filter introduced in line 231 in the original manuscript. Conversely the contributions of traits or catchments is generally distributed among multiple elements of the transformed matrix and the principal components in the PCA, making it difficult to produce clear categories or connect groups of traits to specific hydrological behaviors.

COMMENT #10 Figure 13: what is the unit of MA41?

AUTHOR RESPONSE #10: The streamflow index identified by the code MA41 refers to the "mean annual flow divided by catchment area" and its dimensions are $m^3.s^{-1}.km^{-2}$. We will include units when we refer to the indices if asked to revise the manuscript.

REFERENCES IN OUR RESPONSES:
Addor, N., Nearing, G., Prieto, C., Newman, A. J., Le Vine, N., & Clark, M. P., 2018. A ranking of hydrological signatures based on their predictability in space. Water Resources Research, 54, pp.8792–8812.

Beyer K., Goldstein J., Ramakrishnan R., Shaft U., 1999. When is Nearest Neighbors Meaningful? ICDT Conference Proceedings

Eng, K., Grantham, T.E., Carlisle, D.M. and Wolock, D.M., 2017. Predictability and selection of hydrologic metrics in riverine ecohydrology. Freshwater Science, 36(4), pp.915-926.

Rousseeuw, P.J., 1987. Silhouettes: a graphical aid to the interpretation and validation of cluster analysis. Journal of computational and applied mathematics, 20, pp.53-65.

---

## Author Comment (AC2)

**Manuscript title**
A Network Approach for Multiscale Catchment Classification using Traits
**Authors**
Fabio Ciulla and Charuleka Varadharajan

**RC2**

COMMENT #1 - This article describes the application of a post-PCA clustering algorithm for classification, in this case for catchments. There is no strong argument that the technique is much better than other methods in this particular application, but the breadth, quality and density of the GAGES-II dataset make it an attractive test bed.

The authors do not apply any effort in showing the improvement their technique makes over others. For example, the justification for their network-based approach is a single paragraph and three numbers. In a more structured analysis, the differences between PCA only, and each of the three post-PCA clustering techniques, would be outlined and their differences tabulated with relevant measures (with an equivalent of Figure 3 for each). There would also be a baseline measure, the PCA or one clustering technique with a minimum number of clusters, and some limited exploration of the number of clusters (or the two free parameters mentioned).

AUTHOR RESPONSE #1: We acknowledge that the manuscript we submitted does not provide sufficient evidence that our proposed method based on networks and cosine similarity performs better than traditional unsupervised clustering algorithms. We thank the reviewer for this suggestion and in response performed a more comprehensive analysis comparing the performance of our method against benchmark hierarchical clustering and k-means approaches.

We reiterate that because of the lack of benchmark datasets, it is not straightforward to evaluate the performance of unsupervised methodologies because of the lack of classified target variables. In the absence of a benchmark dataset, we identified two metrics to evaluate the performance of the different methods.

The first metric, which we refer to as the "cluster similarly metric" has been already introduced in the paper (lines 459-472) and reflects the similarity in traits of the catchment clusters. Clusters can be represented by vectors using the average trait z-scores aggregated among the catchments belonging to each cluster. In this way each catchment cluster can be compared with others by calculating the pairwise cosine similarity. For each catchment cluster, the highest value of the similarity is used as a conservative measure of inter cluster similarity, for the purpose of assessing how far apart the clusters are from each other. The median of all the highest similarities represent how distinct the clusters produced by each algorithm are. A good algorithm should produce distinct clusters, so we aim to *minimize this metric*.

The second metric that we now calculated is the silhouette score (Rousseeuw, 1987), which measures how similar each element (i.e., a catchment) is to the cluster it belongs to with respect to the other clusters. The values of this metric range between -1 and 1, with higher values denoting that an element is well placed in its cluster compared to other clusters. The silhouette values are averaged for all the items in the dataset. A good clustering algorithm would produce *higher values of the silhouette score*.

We use these two metrics to compare our clustering approach based on networks and cosine similarity against the hierarchical clustering (in its common implementation using the ward criterion) and the k-means clustering algorithm. Additionally, we also compare our method against a version where the pairwise similarity between nodes is computed using the euclidean distance instead of the cosine distance. This is done to show the difference produced by the metric choice while keeping the rest of the workflow unmodified. Finally, to show the robustness of our approach (and in response to reviewer 2's comment #3), we extended the comparison by exploring the landscape of the two free parameters in our workflow, namely the number of reduced dimensions after the PCA (k) and the cluster granularity. This last quantity is governed by different parameters according to the clustering method used. Three different values of k are investigated; k=6 corresponding to 50% of retained information after PCA, k=20 (our choice in the study) corresponding to 72% of retained information, and k=90 corresponding to 95% of retained information. For each value of k we generated clusters with the different methods so that the number of clusters covering 95% of the dataset ranges between 20 and 120.

The results of this investigation are shown in Fig. 1, which displays the cluster similarity metric, and Fig. 2, showing the silhouette score. According to these two metrics the performance of our network based clustering (red and green points) is considerably superior to both k-means (yellow points) and hierarchical clustering (blue points) across the different values of k and cluster granularity. This is evident from the consistently low values of the median cluster similarity and higher values of silhouette scores. Also, the network generated using the cosine distance as a similarity metric (red points) performs better than its counterpart that uses the euclidean distance (green points). This confirms that the cosine similarity should be preferred as a metric distance in our investigation, where the dimensionality of the problem can be high and the directionality of the data carries valuable information. If invited to resubmit, we would include this additional analysis as part of the "Methods", "Results" and "Discussion" sections.

[Figure]

Figure 1. Median cluster similarity values for different clustering methods and similarity measures used in the network analysis. The number of reduced dimensions after PCA is equal to (a) 6, (b) 20 (used in the paper) and (c) 90 corresponding to 50%, 72% and 95% of retained information respectively. The vertical black dashed line in (b) refers

to the cluster granularity used in the paper. The colored dashed lines are shown for visualization of trends. Lower values of the median cluster similarity metric correspond to better clustering performance.

[Figure]

Figure 2. Silhouette scores for different clustering methods and similarity measures used in the network analysis. The number of reduced dimensions after PCA is equal to (a) 6, (b) 20 (used in the paper) and (c) 90 corresponding to 50%, 72% and 95% of retained information respectively. The vertical black dashed line in (b) refers to the cluster granularity used in the paper. The colored dashed lines are shown for visualization of trends. Higher values of the silhouette scores correspond to better clustering performance.

COMMENT #2 - It is not remarkable (line 579) that a classification method using indices and data from a database (of over 300 measures on over 9000 catchments) specifically designed to described gauged catchments for evaluating streamflow would result in a classification that was related to streamflow measures. It will be no surprise to hydrologists that high rainfall, high elevation, forested catchments behave hydrologically differently to flatter, lower rainfall, cropland areas, or that higher rainfall catchments with lots of urban areas get more flooding. What the results might show however is the bidirectionality such that starting from the stream flow indices we get catchment clusters, and that starting from catchment traits we can get groups of catchments with distinct flow behaviour.

AUTHOR RESPONSE #2: Thanks for this comment. Our original write up was intended to highlight that the approach produces intuitive results that are immediately obvious to all readers. Based on this comment, and those from reviewer #1, we have expanded the analysis to include some additional hydrological insights that can be gained with this methodology.

In particular we expanded our analysis of the examples in Section 4.5. In Figure 2, we show the spearman correlation coefficients ($\rho$) between the streamflow indices (y-axis) and catchment traits (x-axis) aggregated as a median on the trait categories generated in our method that reduces trait redundancy. We find that across the 9067 catchments, mean annual runoff (ma41) is not just positively correlated with traits related to precipitation, but also with the presence of mixed forests $\rho=0.45$) and to a lesser degree evergreen forests ($\rho=0.25$). The ma41 index is also negatively correlated with the pastures and grasslands trait category ($\rho=-.40$).   This highlights the role of vegetation in mediating flows, and is somewhat counterintuitive given that in the absence of management, forested catchments with higher evapotranspiration would be expected to have lower flows compared to grasslands.  As shown in Fig 13 of our paper, the catchments where there tends to be higher mean annual runoff is cluster 7, which is in the Pacific Northwest basin. The fh6 index indicating the mean number of moderate floods per year (>3 times median flows) is not just positively correlated with precipitation traits, but also with traits related to developed areas ($\rho=0.40$), croplands ($\rho=0.32$) and temperature ($\rho=0.43$). The fh6 index is inversely correlated with elevation ($\rho=-0.45$), presence of shrublands ($\rho=-0.38$), evergreen forests ($\rho=-0.28$), coarse soils & groundwater ($\rho=-0.35$). These relationships are consistent across other flood indices. For example, the fh7 index showing the propensity for heavy floods (7 times median flows) similarly has a moderate positive correlation with temperature ($\rho=0.44$) and overland flow ($\rho=0.38$), and a moderate negative correlation with elevation ($\rho=-0.39$) and coarse soils/groundwater ($\rho=-0.43$).  This indicates how flooding is affected by the complex relationships between land use, vegetation, soil infiltration capacity and base flows.

The issue of bidirectionality is interesting but beyond the scope of this paper. We are working on building models to predict hydrological indices using trait clusters, and

understanding the traits of signature-based classification as part of multiple follow-on studies.

COMMENT #3: What would also have been of interest is the places where the flow indices and clusters do not match well. For example, if there are two areas that are low slope, low elevation cropland that have distinctly different baseflow regime, one may be influenced by groundwater discharge or a factor not yet captured, and this would be useful additional data to know or require to be collected.

AUTHOR RESPONSE #3: Thanks for the suggestion and we agree that it is interesting to investigate subsets of catchments within a cluster where the flow indices do not match well. To investigate this aspect, we performed a new analysis that focuses on anomalies in the hydrologic indices within catchment clusters.

[revised manuscript text omitted]

COMMENT #4: The citing of references within the text is inconsistent and non-standard, while many of the listed references do not use capital letters where appropriate in journal names or proceedings.

AUTHOR RESPONSE #4: Thanks for pointing that out. We will correct the references if invited to submit a revised manuscript.

REFERENCES IN OUR RESPONSES:
Olden, J.D. and Poff, N.L., 2003. Redundancy and the choice of hydrologic indices for characterizing streamflow regimes. River research and applications, 19(2), pp.101-121.

---

## Author Response (AR1)

**Manuscript title**
A Network Approach for Multiscale Catchment Classification using Traits
**Authors**
Fabio Ciulla and Charuleka Varadharajan

**Response to Reviewer 1 Comments:**

Major comment

COMMENT #1: The authors introduce a novel method to cluster catchments that is based on traits. The dataset is impressive and the network-based classification is, to my understanding, a relevant and innovative approach in this case. Methods and results are well presented.

AUTHOR RESPONSE #1 - We thank the reviewer for the positive comments. The reviewer brings up important concerns, and we have provided detailed responses below for each concern that was raised. We have made substantial changes that address these comments in the revised manuscript. We also tried to condense some of the original text and merged Figures 6 and 7 of the original manuscript to make room for some of the major additions.

COMMENT #2 - My main concern with such unsupervised classification is how we can use it for practical hydrological studies.

AUTHOR RESPONSE #2: We agree with the comments from both reviewers that we could do more to demonstrate the utility of our methodology for hydrological applications. In addition to responding to comments from reviewer 1 below, we note that we have done additional analysis to show how we can gain hydrological insights from our approach in our response to comments from reviewer 2.

COMMENT #3: From the introduction and discussion, it appears one aim of clustering is application to ungauged basins. In this sense, the results of the paper are discouraging, because the clustering technique does not succeed in relating 'traits' clusters to hydrological behaviors, except for some specific hydrological traits. This part is essential, in my opinion, for switching from a mere clustering exercise to something which could actually be useful in hydrological practice. I do not know how the method can be tuned to improve the overlap between the geographical and hydrological clusters, but my wish is that the authors tackle this issue in the paper. I realize that this implies a significant change in the paper.

AUTHOR RESPONSE #3: As pointed out, one of the applications of our methodology is for predictions in unmonitored basins (PUBs). We had originally demonstrated that our approach to trait-based clustering (which the reviewer refers to as geographical clustering) results in statistically distinct hydrological behavior (based on signatures) relative to other

catchments within the CONUS. We assume the reviewer thinks the results are discouraging, based on the boxplots shown in Figure 13 in the original manuscript, where it appears as though some of the distributions overlap for the two indices shown.

However, we would like to point out that our current approach does result in distinguishing the streamflow behavior for most of the indices, indicating that there already is significant overlap between the trait-based clustering and the hydrological groupings. We made the following changes to address this comment.

First, we added the following text in Section 2.10, line numbers 364-373

[revised manuscript text omitted]

As mentioned in our earlier response, Specifically regarding the statements *"the clustering technique does not succeed in relating 'traits' clusters to hydrological behaviors, except for some specific hydrological traits. This part is essential, in my opinion, for switching from a mere clustering exercise to something which could actually be useful in hydrological practice."* , we note that since the dataset of traits is large, and encompasses a broad set of categories (climate, geology, land use, human activities etc.), it is expected that not all traits (or trait clusters) are going to be related to hydrologic behavior. The question about which traits are most relevant for a particular hydrologic function of interest, such as streamflows, is still largely unresolved. In general, prior large sample studies such as (Eng et al., 2017) and (Addor et al., 2018) have not had much success in using traits to predict hydrologic signatures with statistical or classical machine learning approaches. Tackling the issue of relating trait clusters to hydrologic behavior is a different study that is out of scope for this paper. This is the subject of multiple follow-on studies and papers that we are working on, which require building models to show the relationships between the trait clusters and hydrologic signatures. Please see our response #4 below where we elaborate more on the practical uses for our approach.

COMMENT #4: In the case the authors stick to unsupervised clustering, I guess that the paper might be of interest, but in my opinion, the authors should:

- introduce in more details the practical implications of such clustering, and

AUTHOR RESPONSE #4:

We have added a significant amount of text to describe the practical implications of our workflow as described below. Also see responses #12 and #13 for additional analysis that was added as per reviewer 2's suggestions.

First, we added a new Section 4.7, lines 677-686 with the following text:

*"Our methodology can be used as initial steps of typical workflows used for predictions for unmonitored basins, which is (1) to classify catchments into groups for regionalization, and (2) to select a subset of traits from a large predictor space, a common challenge in many large sample studies. For the former, we choose to classify catchments using traits, since geospatial datasets are now available with a lot of trait information that allows us to do catchment classification at large spatial scales including for unmonitored catchments. For the latter, we find that we can condense a very large dataset containing hundreds of traits into 25 trait categories with the network approach, due to the redundancy in the traits. Thus our paired catchment-trait networks approach not only classifies catchments into clusters, but enables the reduction of a large dataset of traits into an interpretable set of trait categories by eliminating their redundancy. This provides the ability to identify distinct trait*

*categories that are over- or under-expressed in catchment clusters to streamflow behaviors. The parallel analysis of cluster and traits data as networks is an important characteristic that distinguishes our method from other typical unsupervised clustering workflows."*

We added the following text to Section 4.2, lines 532-538

*"To illustrate how our approach helps with reducing trait redundancy, we computed the spearman correlation coefficients ($\rho$) between streamflow indices and catchment traits, which account for non linearities in the data. We find that traits belonging to the same trait categories have similar correlations with the streamflow indices (Fig. I1 in appendix). We find that the spearman correlation coefficients between the streamflow indices and individual catchment traits can be effectively represented as an aggregated median value for the trait categories generated in our method (Fig. 10), which indicates that our reduced set of trait categories are sufficient to determine the general relationships between traits and hydrological signatures. "*

[Figure]

Figure I1. Heatmap showing spearman correlation coefficients between streamflow indices and the traits used in the study. The intensity of the colors show the degree of correlation or anticorrelation as indicated in the color bar. Traits on the horizontal axis are ordered and colored according to the traits categories they belong to. Gray boxes indicate correlations that are above the significance level of p>0.05. The primary purpose of this plot is to provide a visual representation of the redundancy in the correlations between the traits within the same category and streamflow indices.

[Figure]

Figure 10. Heatmap showing spearman correlation coefficients between streamflow indices and the traits categories generated in the study. The intensity of the colors show the degree of correlation or anticorrelation as indicated in the color bar. Gray boxes indicate correlations that are above the significance level of p>0.05.

We also added an additional example for the use of our methodology in Section 4.5, lines 620-637:

*"Additionally, we utilize the ability to determine correlations between stream flow indices and a reduced set of interpretable trait categories (Sect. 4.2). We find that across the 9067 catchments, mean annual runoff (ma41) is not just positively correlated with traits related to precipitation, but also with the presence of mixed forests ($\rho$ = 0.45) and to a lesser degree evergreen forests ($\rho$ = 0.25). The ma41 index is also negatively correlated with the 'pastures and grasslands' trait category ($\rho$ = −.40). This highlights the role of vegetation in mediating flows, and is somewhat counterintuitive given that in the absence of management, forested catchments with higher evapotranspiration would be expected to have lower flows compared to grasslands. Similarly, the fh6 index is not just positively correlated with precipitation traits, but also with traits related to developed areas ($\rho$ = 0.40),*

*croplands (ρ = 0.32) and temperature (ρ = 0.43). It is also inversely correlated with elevation (ρ = −0.45), presence of shrublands (ρ = −0.38), evergreen forests (ρ = −0.28), coarse soils and groundwater (ρ = −0.35). These relationships are consistent across other flood indices. For example, the fh7 index showing the propensity for heavy floods (above 7 times median flows) similarly has a moderate positive correlation with temperature (ρ = 0.44) and overland flow (ρ = 0.38), and a moderate negative correlation with elevation (ρ = −0.39) and coarse soils/groundwater (ρ = −0.43). This indicates how flooding is affected by the complex relationships between land use, vegetation, soil infiltration capacity and base flows.*

*These examples highlight the use of our methodology to demonstrate how specific hydrological behaviors can be connected to catchment traits such as their climatic conditions, topography, land use or anthropogenic influence. The ability to link catchment traits to specific hydrological behaviors, enables further analysis of the factors that influence different stream flow characteristics (e.g. high versus low flows). In particular, the distinction between anthropogenically-influenced trait categories and natural traits (see Table 1) enables further analysis of human activities on hydrologic behavior."*

Hence we argue that the trait categories generated using our method are interpretable in a manner that is harder to do with a dimensionality reduction approach using eigenvectors, where the contributions of traits can be distributed across many principal components. For additional applications of our unsupervised clustering approach, we refer the reviewer to our response #13 to reviewer 2 comments. Here we conducted additional analysis to identify relationships between hydrologic signatures and the catchment clusters, as well as the predominant traits of the catchments in those clusters. This analysis has produced new insights that on its own can be used to generate hypotheses about processes that influence hydrological behavior.

COMMENT #5:

- compare the obtained classification with a benchmark clustering approach.

AUTHOR RESPONSE #5: We acknowledge that the manuscript we submitted does not provide sufficient evidence that our proposed method based on networks and cosine similarity performs better than traditional unsupervised clustering algorithms. We thank the reviewer for this suggestion and in response performed a more comprehensive analysis comparing the performance of our method against benchmark hierarchical clustering and k-means approaches. To address this comment, we made the following changes.

First, we added the following text to methods as a new section 2.11, lines 375-400

[revised manuscript text omitted]

Minor comments

COMMENT #6: l.5: please clarify the term "subject to degradation"

AUTHOR RESPONSE #6: The term "degradation" is often used in computational literature dealing with metrics in high dimensions and refers to the property of a distance metric to perform worse as the number of dimensions grows. This concept of "degradation" is related to the "curse of dimensionality". It can be understood by recognizing that, counter to our intuition, what applies in three dimensions does not necessarily hold in higher dimensions. For example, in high dimensions, most of the mass of the points distributed according to a well behaved Gaussian distribution does not lie around the mean but becomes increasingly distant from it. Most of the mass migrates toward the surface of the domain leaving the bulk of the inner space empty. One of the consequences of this increased sparsity in high dimensional space is that the ratio of the distances of the nearest and farthest neighbors to a given point is almost 1, namely the points become uniformly distant from each other (Beyer et al., 1999).

To clarify, we added the following to the manuscript in Section 4.1, lines 481-484

*"In particular, the distance metrics perform worse (referred to as "degradation") as the number of dimensions grows (Aggarwal et al., 2001). One of the consequences of this phenomenon is that the ratio of the distances of the nearest and farthest neighbors to a given point in high dimensions approaches 1, meaning that the points become uniformly distant from each other (Beyer et al., 1999)."*

COMMENT #7: l.43, l.48 and in many other places: problems with in-line referencing.

AUTHOR RESPONSE #7: Thanks for pointing that out. We have corrected these references.

COMMENT #8: Section 2.3: I understand that traits values are standardized, but are their distributions normal? I guess no and I wonder how this may affect PCA and low dimensional vectors extracted from PCA.

AUTHOR RESPONSE #8: The referee's intuition about the non-normality behavior of the traits distribution is correct. The Shapiro-Wilk test, which checks if the data is drawn from a normal distribution, reveals that none of the traits are normally distributed when testing with a p-value of 0.05. However, the PCA method does not require normality in the input data. However, there are other factors that have to be considered when choosing PCA as the dimensionality reduction approach, which includes whether there are non-linear relationships between the variables, and whether the dataset contains outliers. To address these concerns, we added the following text in Section 2.3, lines 184-192:

*"The traits in the GAGES-II dataset contain significant redundancies, with 84% of pair-wise Pearson correlation coefficients (Pearson, 1895) and 92% of pair-wise Spearman coefficients, which accounts for non-linear relationships (Spearman, 1987), have a significant p-value of 0.05. The coefficient of determination between these two metrics is equal to 0.76, which indicates that although nonlinear relationships among the traits are present, they are not so dominant to prevent the use of a linear dimensionality reduction method such as PCA. Another factor that can affect the PCA algorithm's performance is the presence of outliers. We determined that the PCA is a reasonable choice for the GAGES-II dataset, since only 8.1% of the traits lies outside their "inner fence", a common threshold for outliers, defined as the range between Q1 - 1.5 \* IQR and Q3 + 1.5 \* IQR for each trait, where Q1 and Q3 are the first and third quartiles respectively, and IQR = Q3 - Q1 is the interquartile range. "*

COMMENT #9: l.473-475: Please clarify the added values of the network-based approach compared to other clustering techniques. Many of them address already the problem of dimensionality by working on Eigen-vectors.

AUTHOR RESPONSE #9: In response #4, we explain some of the benefits of our clustering approach. As shown in response #5, we have demonstrated that the network-approach outperforms other clustering techniques. Specific to the comment regarding the use of Eigen-vectors, we point out that although working with eigenvectors reduces the dimensionality of the problem, often the reduced vector space is still high dimensional. In our case, retaining 72% of the information - using the variance explained by the SVD singular values matrix - from the dimensionality reduction, still leads to a 20 dimension vector space. Although there is no universally accepted threshold for "high-dimensional" data, we argue that 20 constitutes a high number of dimensions where distance calculations are impacted.

Using a network approach allows one to choose the similarity metric and not to rely on euclidean distance, a metric needed in most of the traditional unsupervised clustering methods like k-means and hierarchical clustering. Thus we are able to use the cosine similarity metric that is less affected by issues of high dimensionality and include directionality information in the data. This is explained in the Method section 2.11, Result section 3.6, and Discussion section 4.1.

The resulting clusters from our network approach are computed using the information from both the transformed matrix and the principal components. As illustrated in response #4, we believe that our workflow produces more interpretable results than the ones that would be obtained using only the Eigenvectors obtained from a PCA. This is because the network approach allows to separate traits or catchments into distinct clusters, produced using network connections statistics validated by the disparity filter introduced in line 231 in the original manuscript. Conversely the contributions of traits or catchments is generally distributed among multiple elements of the transformed matrix and the principal components in the PCA, making it difficult to produce clear categories or connect groups of traits to specific hydrological behaviors.

COMMENT #10 Figure 13: what is the unit of MA41?

AUTHOR RESPONSE #10: The streamflow index identified by the code MA41 refers to the "mean annual flow divided by catchment area" and its dimensions currently are $m^3.s^{-1}.km^{-2}$. We acknowledge that this is an uncommon choice and have changed into the more commonly used mm/day. Units for MA41 are explicitly indicated in Figure 14 axis and caption of the revised manuscript.

**Response to Reviewer 2 Comments:**

COMMENT #11 - This article describes the application of a post-PCA clustering algorithm for classification, in this case for catchments. There is no strong argument that the technique is much better than other methods in this particular application, but the breadth, quality and density of the GAGES-II dataset make it an attractive test bed.

The authors do not apply any effort in showing the improvement their technique makes over others. For example, the justification for their network-based approach is a single paragraph and three numbers. In a more structured analysis, the differences between PCA only, and each of the three post-PCA clustering techniques, would be outlined and their differences tabulated with relevant measures (with an equivalent of Figure 3 for each). There would also be a baseline measure, the PCA or one clustering technique with a minimum number of clusters, and some limited exploration of the number of clusters (or the two free parameters mentioned).

AUTHOR RESPONSE #11: We acknowledge that the manuscript we submitted does not provide sufficient evidence that our proposed method based on networks and cosine similarity performs better than traditional unsupervised clustering algorithms. We thank the reviewer for this suggestion. See response #5 where we performed a more comprehensive analysis comparing the performance of our method against benchmark hierarchical clustering and k-means approaches, as well as exploration of the free parameters.

COMMENT #12 - It is not remarkable (line 579) that a classification method using indices and data from a database (of over 300 measures on over 9000 catchments) specifically designed to described gauged catchments for evaluating streamflow would result in a classification that was related to streamflow measures. It will be no surprise to hydrologists that high rainfall, high elevation, forested catchments behave hydrologically differently to flatter, lower rainfall, cropland areas, or that higher rainfall catchments with lots of urban areas get more flooding. What the results might show however is the bidirectionality such that starting from the stream flow indices we get catchment clusters, and that starting from catchment traits we can get groups of catchments with distinct flow behavior.

AUTHOR RESPONSE #12: Thanks for this comment. Our original write up was intended to highlight that the approach produces intuitive results that are immediately obvious to all readers. Based on this comment, and those from reviewer #1, we have expanded the analysis to include some additional hydrological insights that can be gained with this methodology. In particular we expanded our discussion Section 4.2 and Section 4.5 (see response #4 for changes made to the manuscript). We also removed the text referring to our results as remarkable.

The issue of bidirectionality is interesting but beyond the scope of this paper. We are working on building models to predict hydrological indices using trait clusters, and understanding the traits of signature-based classification as part of multiple follow-on studies.

COMMENT #13: What would also have been of interest is the places where the flow indices and clusters do not match well. For example, if there are two areas that are low slope, low elevation cropland that have distinctly different baseflow regime, one may be influenced by groundwater discharge or a factor not yet captured, and this would be useful additional data to know or require to be collected.

AUTHOR RESPONSE #13: Thanks for the suggestion and we agree that it is interesting to investigate subsets of catchments within a cluster where the flow indices do not match well. To investigate this aspect, we performed a new analysis that focuses on anomalies in the hydrologic indices within catchment clusters. We have added the following text to the discussion as a new Section 4.6 "Examining diversity of hydrologic behavior within catchment clusters" in lines 639-689:

[revised manuscript text omitted]

COMMENT #14: The citing of references within the text is inconsistent and non-standard, while many of the listed references do not use capital letters where appropriate in journal names or proceedings.

AUTHOR RESPONSE #14: Thanks for pointing that out. We have corrected the references.

REFERENCES IN OUR RESPONSES:
Addor, N., Nearing, G., Prieto, C., Newman, A. J., Le Vine, N., & Clark, M. P., 2018. A ranking of hydrological signatures based on their predictability in space. Water Resources Research, 54, pp.8792–8812.

Beyer K., Goldstein J., Ramakrishnan R., Shaft U., 1999. When is Nearest Neighbors Meaningful? ICDT Conference Proceedings

Eng, K., Grantham, T.E., Carlisle, D.M. and Wolock, D.M., 2017. Predictability and selection of hydrologic metrics in riverine ecohydrology. Freshwater Science, 36(4), pp.915-926.

Rousseeuw, P.J., 1987. Silhouettes: a graphical aid to the interpretation and validation of cluster analysis. Journal of computational and applied mathematics, 20, pp.53-65.

Olden, J.D. and Poff, N.L., 2003. Redundancy and the choice of hydrologic indices for characterizing streamflow regimes. River research and applications, 19(2), pp.101-121.